# Learning To Learn Around A Common Mean

**Giulia Denevi**[1,2], **Carlo Ciliberto**[3,4], **Dimitris Stamos**[4] **and Massimiliano Pontil**[1,4]
[1]Istituto Italiano di Tecnologia (Italy), [2]University of Genoa (Italy),
[3]Imperial College of London (UK), [4]University College of London (UK)

## Abstract

The problem of learning-to-learn (LTL) or meta-learning is gaining increasing attention due to recent empirical evidence of its effectiveness in applications. The goal addressed in LTL is to select an algorithm that works well on tasks sampled from a meta-distribution. In this work, we consider the family of algorithms given by a variant of Ridge Regression, in which the regularizer is the square distance to an unknown mean vector. We show that, in this setting, the LTL problem can be reformulated as a Least Squares (LS) problem and we exploit a novel meta-algorithm to efficiently solve it. At each iteration the meta-algorithm processes only one dataset. Specifically, it firstly estimates the stochastic LS objective function, by splitting this dataset into two subsets used to train and test the inner algorithm, respectively. Secondly, it performs a stochastic gradient step with the estimated value. Under specific assumptions, we present a bound for the generalization error of our meta-algorithm, which suggests the right splitting parameter to choose. When the hyper-parameters of the problem are fixed, this bound is consistent as the number of tasks grows, even if the sample size is kept constant. Preliminary experiments confirm our theoretical findings, highlighting the advantage of our approach, with respect to independent task learning.

## 1 Introduction

Learning-to-learn (LTL) or meta-learning addresses the problem of learning an algorithm that "works well" on a class of learning tasks, which are randomly observed via a corresponding finite set of training examples, see [4, 19, 29] and references therein. The learning tasks are assumed to share specific similarities, in that they are sampled from a common meta-distribution, often referred to as *environment* in literature [4]. The LTL process aims at leveraging such similarities in order to learn an algorithm which is well suited to learn new tasks sampled from the same environment. This approach brings a substantial improvement over learning the tasks in isolation – known as independent task learning (ITL) – especially when the sample size per task is small, a common setting in many applications [6, 27, 30]. These ideas are strongly related to multitask learning (MTL) [7, 8, 9, 10, 14, 18, 23]. The key difference is that in MTL the goal is to perform well on the observed tasks while in LTL the aim is to perform well on "future" tasks.

In this work, we study a particular kind of environment, in which the randomly observed tasks are linear regression problems and the underlying family of learning algorithms is *Ridge Regression around a common mean*, that is, Ridge Regression in which we introduce in the regularizer a bias term, playing the role of a common mean among the tasks. Starting from a stream of datasets sampled from this environment, our goal is to learn the common mean by minimizing the *transfer risk* [4, 19], which measures the average error of the underlying learning algorithm, trained on a random dataset from the environment. Although previous theoretical investigations of LTL minimize a proxy for the transfer risk, given by the average multitask empirical error [4, 19, 20, 22, 24] or the so-called *future empirical risk* [12] of the algorithm, as a first contribution of this work, we show that the specific family of algorithms considered here, naturally lends itself to directly minimize the transfer risk of

the algorithm trained with less points. More precisely, we first observe that, in this setting, the LTL problem can be reformulated as a Least Squares (LS) problem, the structure of which depends on the environment. After this, in order to compute it, motivated by recent empirical studies on few shot-learning and meta-learning [15, 16, 26], we split the datasets of the training tasks in a subset used to train the algorithm and a subset used to estimate its risk.

LTL is particularly appealing when considered from an online or incremental perspective, in which we receive in input a sequence of datasets and the goal is to efficiently update an underlying learning algorithm, which will then be applied to the next yet-to-be-encountered task, without the need to keep in memory previously encountered datasets. Our second contribution is to show that the specific LTL problem studied in this paper can be naturally tackled incrementally by Stochastic Gradient Descent (SGD) procedures and, when the environment satisfies certain assumptions given in the paper, we provide a complete statistical analysis of this approach, which highlights the role of the splitting parameter, namely, the number of points we use to train the inner algorithm. Moreover, in such a case, when the hyper-parameters of the problem are fixed, a remarkable feature of our learning approach is that it provides a consistent estimate of the transfer risk as the number of tasks grows, even if the tasks' sample size is kept constant, whereas classical approaches would need the sample size to grow as well. Our proof technique leverages previous work on stochastic optimization for LS [13] with tools from classical LTL theory [4, 19, 20, 22].

**Paper organization.** In Sec. 2 we introduce the LTL problem for the class of algorithms based on Ridge Regression around a common mean and we show that this problem is equivalent to a LS problem. In Sec. 3 we describe the online approach through which we directly attempt to minimize the transfer risk for this family of learning algorithms. In Sec. 4 we provide the statistical analysis for this approach, under specific assumptions on the environment. Sec. 5 presents preliminary experiments confirming our theoretical observations and, finally, in Sec. 6, we draw our conclusion and we highlight possible future directions. All the technical proofs are reported in the appendix.

**Previous work.** Although LTL is naturally suited for the incremental setting, we are only aware of few theoretical investigations about it [1, 3, 12, 17, 25]. Most related to our work are [1], where the authors consider a general PAC-Bayesian approach to LTL and [12], which considers the problem of learning a linear representation shared by the tasks. However, both these papers address a different class of learning algorithms. Furthermore, the approach presented here follows a different strategy of directly minimizing the transfer risk. In literature, the LTL problem has been almost exclusively considered in the setting in which the tasks are given in one batch [4, 19, 20, 21, 22, 24], as opposed to sequentially. Perhaps most related to our work is the paper [24], where the authors consider the same family of learning algorithms analyzed in this paper, but in a PAC-Bayesian setting.

## 2 Learning-to-Learn Problem

In this work we focus on the standard linear regression setting. Let $\mathcal{Z} = \mathcal{X} \times \mathcal{Y}$ be the data space, where $\mathcal{X} \subseteq \mathbb{R}^d$ is the input space and $\mathcal{Y} \subseteq \mathbb{R}$ the output space. We denote by $x, y, z$ generic elements of $\mathcal{X}, \mathcal{Y}, \mathcal{Z}$, respectively, where $z = (x, y)$. The discrepancy between two outputs $y, y' \in \mathcal{Y}$ is measured by the square loss $\ell(y, y') = \frac{1}{2}(y - y')^2$. The symbols $\|\cdot\|$ and $\langle \cdot, \cdot \rangle$ denote the standard norm and the standard scalar product in the Euclidean space and $\cdot^\top$ represents the standard transpose operation. We also let $\mathcal{B}_1 \subseteq \mathbb{R}^d$ be the zero-centered unit ball in $\mathbb{R}^d$. For a real matrix $M$, we denote by $\|M\|_\infty$ its spectral (operator) norm and by $\|M\|$ its Frobenius norm. We let $\lambda_{\max}(M) = \|M\|_\infty$ and $\lambda_{\min}(M)$ be the largest and smallest eigenvalue of a symmetric positive semidefinite (PSD) real matrix $M$, respectively. The symbols $\preceq$ and $\prec$ denote the classical ordering among real symmetric matrices and, for any positive integer $k \in \mathbb{N}$, we define the set of integer numbers $[k] = \{1, \ldots, k\}$. Finally, all the expectations must be intended accordingly to the context.

### 2.1 Environment and Transfer Risk

We consider the setting where we sequentially receive a stream of datasets $Z_n^{(1)}, \ldots, Z_n^{(T)}, \ldots$, sampled from a fixed environment $\rho$, each of which is formed by $n$ points[1]. Starting from these

datasets, we wish to *learn a learning algorithm* (hence the name LTL) that performs well on a new random task sampled from the environment. Specifically, we wish to find an *inner algorithm* $A : \cup_{n \in \mathbb{N}} (\mathcal{X} \times \mathcal{Y})^n \to \mathbb{R}^d$, such that, when we train it with a new dataset *composed by $n$ points*[2] and sampled from the environment, the corresponding error is low. This objective translates into requiring that the *transfer risk* of the algorithm $A$ trained with $n$ points over the environment $\rho$, defined as

$$\mathcal{E}_n(A) = \mathbb{E}_{\mu \sim \rho} \mathbb{E}_{Z_n \sim \mu^n} \mathbb{E}_{z \sim \mu} \frac{1}{2} \Big[ \big( \langle x, A(Z_n) \rangle - y \big)^2 \Big] \tag{1}$$

is as small as possible. This quantity measures the expected error (risk) that the algorithm $A$, trained on the dataset $Z_n$, incurs *on average with respect to the distribution of tasks $\mu$ induced by $\rho$*. That is, to compute the transfer risk, we first draw a task $\mu \sim \rho$ and a corresponding $n$-sample $Z_n \in \mathcal{Z}^n$ from $\mu^n$, we then apply the learning algorithm to obtain the estimator $A(Z_n)$ and finally we measure the risk of this estimator on the distribution $\mu$ by taking the expectation over a test point $z$.

Throughout the paper, we make the following assumption.

**Assumption 1** (Linear regression tasks). *The meta-distribution $\rho$ samples linear regression tasks parametrized as $\mu = (w, p, \eta)$, where $w \in \mathbb{R}^d$ is the regression vector, $p$ is the input marginal and $\eta$ is the noise distribution such that $\eta \mid p$ has zero-mean, for almost every $p$. In particular, the sampling of a datapoint from $\mu$ must be intended as*

$$(x, y) \sim \mu \iff x \sim p, \quad \epsilon \sim \eta, \quad y = \langle x, w \rangle + \epsilon. \tag{2}$$

*Moreover, denoting by $\Sigma_\mu = \mathbb{E}_{z \sim \mu} x x^\top$, we assume that $\mathbb{E}_{\mu \sim \rho} \lambda_{\min}(\Sigma_\mu) > 0$*[3].

In this work, each dataset $Z_n$ is given by an i.i.d. sample of $n$ points $(x_i, y_i)_{i=1}^n \sim \mu^n$. In the following, we will use the more compact notation $Z_n = (X_n, \mathbf{y}_n)$, where $X_n \in \mathbb{R}^{n \times d}$ denotes the matrix having the vectors $x_i$ as rows, $\mathbf{y}_n = (y_i)_{i=1}^n \in \mathbb{R}^n$ is the vector of labels and $\boldsymbol{\epsilon}_n = (\epsilon_i)_{i=1}^n \in \mathbb{R}^n$ is the vector containing the noise on the labels. Moreover, by Asm. 1 the model equation $\mathbf{y}_n = X_n w + \boldsymbol{\epsilon}_n$ holds. We will denote by $z = (x, y) \sim \mu$ a test point which must be always intended to be independent from the training set, and in an analogous way, we will have $y = \langle x, w \rangle + \epsilon$. The above assumption on the environment is mainly made to simplify our exposition; the general case may be pursued by considering the approximation error due to the choice of the class of the linear functions and a possible dependency of the noise on the inputs (heteroscedastic noise).

## 2.2 The Family of Learning Algorithms

The subject of our study in this paper is the family of learning algorithms based on Ridge Regression, where we introduce a further bias term in the regularizer. More precisely, for any training dataset $Z_n = (X_n, \mathbf{y}_n) \sim \mu^n$ and for any $h \in \mathbb{R}^d$, we consider the following algorithm

$$Z_n \mapsto A(Z_n) \equiv w_h(Z_n) = \underset{w \in \mathbb{R}^d}{\operatorname{argmin}} \frac{1}{n} \|X_n w - \mathbf{y}_n\|^2 + \lambda \|w - h\|^2, \tag{3}$$

where $\lambda > 0$ is a hyper-parameter of the algorithm. A direct computation gives the closed form

$$w_h(Z_n) = C_{\lambda,n}^{-1} \Big( \frac{X_n^\top \mathbf{y}_n}{n} + \lambda h \Big), \quad C_{\lambda,n} = \frac{X_n^\top X_n}{n} + \lambda I. \tag{4}$$

Our goal is to leverage similarities between the tasks in the environment via the common mean $h$. Intuitively, if the regression vectors $w$ sampled from the environment have a large mean $\bar{w}$ and a small variance, then, applying Ridge Regression around $\bar{w}$ should bring better estimation risk of these regression vectors than solving each task independently, for instance, with standard Ridge Regression ($h = 0$). In this work, we address the problem of choosing an algorithm in the above family that performs well on tasks sampled from the environment according to the above LTL setting. This translates into selecting a parameter $h$ such that the associated algorithm in the family has a small transfer risk. Hence, considering $\lambda$ as a hyper-parameter external to our LTL problem and using the notation $\mathcal{E}_n(h) \equiv \mathcal{E}_n(w_h)$ for the transfer risk in Eq. (1), our goal is to minimize $\mathcal{E}_n$ with respect to $h$. In the next section, we will see that, thanks to the fact that the algorithm is an affine transformation of the parameter $h$ (see Eq. (4)), not only this function is convex (an unusual property in the LTL setting), but it presents also a particular LS structure.

### 2.3 Minimizing the Transfer Risk: a Least Squares Problem

The following proposition establishes that the problem of minimizing $\mathcal{E}_n$ over the common mean $h$ can be reformulated as a LS problem with transformed inputs and outputs, sampled from a distribution induced by sampling an $n$-dataset from the original environment. The proof is reported in App. B.

**Proposition 1** (LS Problem Around a Common Mean for $\mathcal{E}_n$). *For any $\lambda > 0$ and $h \in \mathbb{R}^d$, the transfer risk $\mathcal{E}_n$ in Eq. (1) of the learning algorithm $w_h$ in Eqs. (3)-(4), can be rewritten as*

$$\mathcal{E}_n(h) = \frac{1}{2}\mathbb{E}_{\bar{x}_n, \bar{y}_n}\Big[\big(\langle \bar{x}_n, h \rangle - \bar{y}_n\big)^2\Big] \tag{5}$$

*where the meta-data are given by*

$$\bar{x}_n = \lambda C_{\lambda,n}^{-1}x, \quad \text{and} \quad \bar{y}_n = y - \Big\langle \bar{x}_n, \frac{X_n^\top \mathbf{y}_n}{\lambda n} \Big\rangle.$$

*Moreover, under Asm. 1, the meta-covariance matrix $\bar{\Sigma}_n = \mathbb{E}\bar{x}_n\bar{x}_n^\top = \lambda^2 \mathbb{E}\big[C_{\lambda,n}^{-1}\Sigma_\mu C_{\lambda,n}^{-1}\big]$ is invertible and $h_n^* = \bar{\Sigma}_n^{-1}\mathbb{E}[\bar{y}_n\bar{x}_n]$ is the unique minimizer of the LS function in Eq. (5). In such a case, letting $v = w - h_n^*$, we have that $\bar{y}_n = \langle \bar{x}_n, h_n^* \rangle + \bar{\epsilon}_n$, with $\bar{\epsilon}_n = \epsilon + \Big\langle \bar{x}_n, v - \frac{X_n^\top \boldsymbol{\epsilon}_n}{\lambda n} \Big\rangle.$*

**Remark 1** (A Misspecified LS Problem with Heteroscedastic Noise for $\mathcal{E}_n$). *From Prop. 1 we observe that, even though the original tasks are well-specified linear models with homoscedastic (i.e. not depending on the inputs) noise, without further assumptions on the environment, the linear model for the meta-LS problem is usually not well-specified, that is, in general, it may hold that $\mathbb{E}[\bar{\epsilon}_n|\bar{x}_n] \neq 0$, and, moreover, the noise is heteroscedastic.*

As a consequence of the above proposition, we can exploit standard results about LS. For instance, it is easy to show that, for a generic vector $h \in \mathbb{R}^d$, the *excess transfer risk* $\mathcal{E}_n(h) - \mathcal{E}_n(h_n^*)$ coincides with the weighted square norm $\frac{1}{2}\|\bar{\Sigma}_n^{1/2}(h - h_n^*)\|^2$. In particular, the gap between the performance of standard Ridge Regression ($h = 0$), corresponding to solving each task independently, and the best algorithm in our class (i.e. the algorithm associated to the parameter $h_n^*$) is given by $\mathcal{E}_n(0) - \mathcal{E}_n(h_n^*) = \frac{1}{2}\|\bar{\Sigma}_n^{1/2}h_n^*\|^2$. Characterizing situations in which this gap is significant is not an obvious point, in the folllowing we will see that, making further assumptions on the environment, we will be able to answer this question (see Rem. 3 below).

### 2.4 Two Examples of Environments

So far we have described the basic requirements of the environment considered in this work, however, making further assumptions about the data generation process will allow us to further analyze the LTL problem. We now discuss two specific examples. Their proofs are reported in App. B.

**Example 1.** *Let $\mathcal{X} \subseteq \mathcal{B}_1$ and let the environment $\rho$ satisfy Asm. 1. Furthermore, assume for almost every $p$ that: i) $\eta \mid p$ has variance bounded by $\sigma_\epsilon^2$, for $\sigma_\epsilon \geq 0$, ii) $\mathbb{E}[w|p] = \bar{w}$, and iii) $\mathbb{E}\big[(w - \bar{w})(w - \bar{w})^\top|p\big] \preceq \sigma_w^2 I$, for $\sigma_w \geq 0$. Then, for any $\lambda > 0$ and $h \in \mathbb{R}^d$, $h_n^* = \bar{w}$ and, letting $A_n = C_{\lambda,n}^{-1}xx^\top C_{\lambda,n}^{-1}$, we have*

$$\mathcal{E}_n(h) \leq \frac{1}{2}\big\|\bar{\Sigma}_n^{1/2}(h - \bar{w})\big\|^2 + \frac{\sigma_w^2}{2}\mathrm{tr}(\bar{\Sigma}_n) + \frac{\sigma_\epsilon^2}{2}\Big(1 + \mathrm{tr}\Big(\mathbb{E}\Big[\frac{X_n^\top X_n A_n}{n^2}\Big]\Big)\Big). \tag{6}$$

The fact that the minimizer of the transfer risk in Ex. 1 does not depend on $n$ will be fundamental in the subsequent analysis and we remark that the proof of this statement exploits only Asm. 1 and the point $ii)$. Moreover, the upper bound in Eq. (6) is tight. For a discussion on the link between our LTL problem for the setting in Ex. 1 and the Mean Estimation problem (see e.g. [11]) we refer to Rem. 9 in App. B. We now point out that Ex. 1 is an exception to what observed in Rem. 1.

**Remark 2** (A Well-Specified LS Problem for Ex. 1). *For the environment in Ex. 1, the linear model for the LS problem described in Prop. 1 is well-specified. Indeed, we have that*

$$\mathbb{E}\big[\bar{\epsilon}_n|\bar{x}_n\big] = \mathbb{E}[\epsilon|p] + \big\langle \bar{x}_n, \mathbb{E}[v|p] - (n\lambda)^{-1}X_n^\top \mathbb{E}[\boldsymbol{\epsilon}_n|p]\big\rangle = 0.$$

*Specifically, to get the above statement, according to Asm. 1, we have exploited the independence of the sampling of $x, w, \epsilon$, conditioned with respect to $p$, the fact that the linear model is well-specified for the original tasks and the relation $\mathbb{E}[v|p] = \mathbb{E}[w|p] - \bar{w} = 0$, which holds in the setting of Ex. 1.*

In the setting of Ex. 1 we can give conditions under which the gap between the performance of ITL and Ridge Regression with the best mean is significant. We state this in the following remark, the proof of which is reported in App. B.

**Remark 3** (Advantage of Learning Around the Best Mean over ITL in Ex. 1). *Consider the setting of Ex. 1. If the noise satisfies $\sigma_\epsilon^2 \ll \left(n^{-1}\lambda^{-2} + 1\right)^{-1}\|\bar{\Sigma}_n^{1/2}\bar{w}\|^2$ and the regression vectors are such that $\sigma_w^2 \ll \operatorname{tr}(\bar{\Sigma}_n)^{-1}\|\bar{\Sigma}_n^{1/2}\bar{w}\|^2$, then $\mathcal{E}_n(0) - \mathcal{E}_n(\bar{w}) \gg \mathcal{E}_n(\bar{w})$.*

A special case of Ex. 1 is when the input marginal is always the same. The next example generalizes this to a mixture and provides a scenario in which the minimizer $h_n^*$ of the transfer risk varies with $n$.

**Example 2.** *Let $\mathcal{X} \subseteq \mathcal{B}_1$ and consider the environment $\rho$ formed by $K \in \mathbb{N}\backslash\{0\}$ clusters of tasks parametrized by the triplet $(w, p, \eta)$ as in Asm. 1. Assume that each cluster $k \in [K]$ is associated to a marginal distribution $p_k$ that is sampled with probability $\mathbb{P}(p = p_k) = \nu_k > 0$. For any $k \in [K]$ and $\lambda > 0$, let $\bar{w}_k = \mathbb{E}[w|p = p_k]$, $\bar{\Sigma}_{n,k} = (n\lambda)^2\mathbb{E}[A_n|p = p_k]$ with $A_n = C_{\lambda,n}^{-1}xx^\top C_{\lambda,n}^{-1}$ and assume that i) $\eta \mid p = p_k$ has variance bounded by $\sigma_{\epsilon,k}^2$ for $\sigma_{\epsilon,k} \geq 0$, ii) $\mathbb{E}[(w - \bar{w}_k)(w - \bar{w}_k)^\top|p = p_k] \preceq \sigma_{w,k}^2 I$ for $\sigma_{w,k} \geq 0$. Then, for any $\lambda > 0$ and $h \in \mathbb{R}^d$, $h_n^* = (\sum_{k=1}^K \nu_k\bar{\Sigma}_{n,k})^{-1}\sum_{k=1}^K \nu_k\bar{\Sigma}_{n,k}\bar{w}_k$ and $\mathcal{E}_n(h) = \sum_{k=1}^K \nu_k\mathcal{E}_{n,k}(h)$, where*

$$\mathcal{E}_{n,k}(h) \leq \frac{1}{2}\big\|\bar{\Sigma}_{n,k}^{1/2}\big(h - \bar{w}_k\big)\big\|^2 + \frac{\sigma_{w,k}^2}{2}\operatorname{tr}\big(\bar{\Sigma}_{n,k}\big) + \frac{\sigma_{\epsilon,k}^2}{2}\Big(1 + \operatorname{tr}\Big(\mathbb{E}\Big[\frac{X_n^\top X_n A_n}{n^2}\Big|p = p_k\Big]\Big)\Big).$$

In the subsequent analysis we will give theoretical guarantees for our LTL approach for Ex. 1 only. However, it is natural to expect that, also in the setting described in Ex. 2, when both the variance within each cluster and the variance between the clusters themselves are small, there should be an advantage in applying our LTL method around a single mean, in comparison to solving each task independently. The experiments in Sec. 5 will confirm this. In the next section, we describe the online meta-algorithm that we propose in order to address the LTL problem outlined above.

## 3 The Splitting Stochastic Meta-Algorithm

We recall that our goal is, given a sequence of datasets $Z_n^{(1)}, \ldots, Z_n^{(T)}$ sampled from $\rho$, to find a parameter $h$ so that the associated inner algorithm in Eqs. (3)-(4) works well on new datasets *of $n$ points* sampled from the same environment. This can be translated into minimizing the transfer risk $\mathcal{E}_n$ in Eqs. (1)-(5). However, in our setting, we cannot directly minimize this function, as we would need a further test point to compute the risk of the inner algorithm. Hence, we proceed as follows.

### 3.1 The Splitting Step

Inspired by recent work on few shot-learning [15, 16, 26], we do not use all the $n$ points in each dataset to train the inner algorithm (i.e. we do not work on what in literature is called *future empirical risk* [12, 20]), but we sacrifice a subset of them for testing. More precisely, we fix a value $r \in [n-1]$ and, when we receive a new dataset $Z_n$, we split it into two parts, $Z_n = (Z_r, Z_{n-r})$, where $Z_r = (X_r, \mathbf{y}_r)$ contains the first $r$ points of $Z_n$ and $Z_{n-r} = (X_{n-r}, \mathbf{y}_{n-r})$ contains the remaining $n - r$ points. Note that the two datasets $Z_r$ and $Z_{n-r}$ are independent one of another, conditioned with respect to the task. Once this splitting is performed, we use $Z_r$ to train the inner algorithm in Eqs. (3)-(4) replacing in its functional form $Z_n$ by $Z_r$, while the remaining part of data $Z_{n-r}$ is used to estimate the transfer risk of the corresponding algorithm by the formula

$$\mathcal{E}_r(h) = \frac{1}{2}\mathbb{E}_{\mu\sim\rho}\mathbb{E}_{Z_r\sim\mu^r}\mathbb{E}_{Z_{n-r}\sim\mu^{n-r}}\frac{1}{n-r}\Big[\big\|X_{n-r}w_h(Z_r) - \mathbf{y}_{n-r}\big\|^2\Big]. \tag{7}$$

Some remarks about the definition of the inner algorithm and Eq. (7) are in order.

**Remark 4** (Normalization Factor). *As described before, for any $r \in [n-1]$, the inner algorithm in Eq. (3) is applied to the dataset $Z_r$, with the same normalization factor $1/n$ (and not $1/r$), i.e*

$$w_h(Z_r) = \underset{w\in\mathbb{R}^d}{\operatorname{argmin}} \frac{1}{n}\big\|X_r w - \mathbf{y}_r\big\|^2 + \lambda\|w - h\|^2 = \underset{w\in\mathbb{R}^d}{\operatorname{argmin}} \frac{1}{r}\big\|X_r w - \mathbf{y}_r\big\|^2 + \frac{\lambda n}{r}\|w - h\|^2. \tag{8}$$

*According to this definition, we are using the biased Ridge Regression with the standard normalization factor, in which we divide the regularization parameter $\lambda$ by the fraction of points used for training. Note that the less the training points, the stronger the effect of the regularization. In such a case the output of the algorithm will be encouraged to stay closer to the estimated mean $h$, thereby transferring more knowledge among the tasks.*

**Remark 5** (Conditional Mini-Batch). *Since the test points in $Z_{n-r}$ are conditional i.i.d. with respect to the training points $Z_r$, the definition of $\mathcal{E}_r$ with more test points in Eq. (7) is equivalent to the one with just one test point. However, from an algorithmic point of view, considering more than one test point in the definition of $\mathcal{E}_r$, will be an important aspect. In fact, even if this technique cannot be properly interpreted as a standard mini-batch – since the test points are not independent with respect to the global distribution (they are just* conditionally *independent) – we will see that, in the setting described in Ex. 1, working with more test points brings similar benefits as standard mini-batches. More precisely, it will reduce the variance of the unbiased estimates of the gradient used by our stochastic approach (see Lemma 19 in App. D). We stress that, in our analysis, the above statement derives from both the specific characteristics of Ex. 1 and the normalization factor $1/n$ in the algorithm.*

In analogy with the case analyzed before for $\mathcal{E}_n$ in Prop. 1, exploiting again the following closed form of the algorithm in Eq. (8)

$$w_h(Z_r) = C_{\lambda,r}^{-1}\Big(\frac{X_r^\top \mathbf{y}_r}{n} + \lambda h\Big), \quad C_{\lambda,r} = \frac{X_r^\top X_r}{n} + \lambda I, \tag{9}$$

we can rewrite the transfer risk $\mathcal{E}_r$ in Eq. (7) as a LS function, but, differently from the setting in Prop. 1, the LS function in this case is vector-valued. The following proposition formalize this. Its proof follows along the same lines of Prop. 1 and it is reported in App. B.

**Proposition 2** (LS Problem Around a Common Mean for $\mathcal{E}_r$). *For any $\lambda > 0$, $h \in \mathbb{R}^d$ and $r \in [n-1]$, the transfer risk $\mathcal{E}_r$ in Eq. (7) of the learning algorithm $w_h$ in Eqs. (8)-(9), can be rewritten as*

$$\mathcal{E}_r(h) = \frac{1}{2}\mathbb{E}_{\bar{X}_r,\bar{\mathbf{y}}_r}\left[\left\|\bar{X}_r h - \bar{\mathbf{y}}_r\right\|^2\right] \tag{10}$$

*where the meta-data are given by*

$$\bar{X}_r = \frac{\lambda}{\sqrt{n-r}}X_{n-r}C_{\lambda,r}^{-1}, \qquad \bar{\mathbf{y}}_r = \frac{1}{\sqrt{n-r}}\Big(\mathbf{y}_{n-r} - \frac{\sqrt{n-r}}{\lambda}\bar{X}_r\frac{X_r^\top \mathbf{y}_r}{n}\Big). \tag{11}$$

*Moreover, under Asm. 1, the meta-covariance matrix $\bar{\Sigma}_r = \mathbb{E}\bar{X}_r^\top \bar{X}_r = \lambda^2\mathbb{E}\big[C_{\lambda,r}^{-1}\Sigma_\mu C_{\lambda,r}^{-1}\big]$ is invertible and $h_r^* = \bar{\Sigma}_r^{-1}\mathbb{E}[\bar{X}_r^\top \bar{\mathbf{y}}_r]$ is the unique minimizer of the LS function in Eq. (10). In such a case, letting $v = w - h_r^*$, we have that $\bar{\mathbf{y}}_r = \bar{X}_r h_r^* + \bar{\boldsymbol{\epsilon}}_r$, with*

$$\bar{\boldsymbol{\epsilon}}_r = \frac{1}{\sqrt{n-r}}\boldsymbol{\epsilon}_{n-r} + \bar{X}_r\Big(v - \frac{X_r^\top \boldsymbol{\epsilon}_r}{\lambda n}\Big). \tag{12}$$

Notice that, in virtue of Rem. 4 and Rem. 5, all the statements in Ex. 1 and Ex. 2 can be extended to $\mathcal{E}_r$ by a rescaling of the parameter $\lambda$. We now point out some observations regarding the LS problem introduced in Prop. 2.

**Remark 6** (A Misspecified Vector-Valued LS Problem with Heteroscedastic Noise for $\mathcal{E}_r$). *Eq. (12) implies that the transformed data points, i.e. the rows of $\bar{X}_r$ and the components of $\bar{\mathbf{y}}_r$ are not independent. Indeed, due to the common training dataset $Z_r$, there are dependencies between the components of the meta-noise in Eq. (12). Similarly to what observed in Rem. 1 for $\mathcal{E}_n$, also the noise on the LS problem associated to $\mathcal{E}_r$ is heteroscedastic, moreover the linear model is usually not well-specified, with the exception of the setting in Ex. 1 (as already observed for $\mathcal{E}_n$ in Rem. 2 ).*

We conclude this section by describing how the previous setting can be naturally extended to the case in which we use all the points in each dataset to test the inner algorithm.

**Remark 7** (The Case $r = 0$). *Interpreting $Z_0 = (X_0, \mathbf{y}_0)$ as the empty set and the associated Tikhonov matrix as $C_{\lambda,0} = \lambda I$, all the results stated above can be extended to the case $r = 0$. Specifically, we define the algorithm as $w_h(Z_0) = \mathrm{argmin}_{w\in\mathbb{R}^d}\left\|w - h\right\|^2 = h$ and its transfer risk as $\mathcal{E}_0(h) = \frac{1}{2}\mathbb{E}_{\mu\sim\rho}\mathbb{E}_{Z_n\sim\mu^n}\left[\frac{1}{n}\left\|X_n h - \mathbf{y}_n\right\|^2\right]$. Again, the points in $Z_n$ are i.i.d., but we keep all of them because of what observed in Rem. 5. Moreover, making the identifications $\bar{X}_r = X_n/\sqrt{n}$ and $\bar{\mathbf{y}}_r = \mathbf{y}_n/\sqrt{n}$, the statements in Prop. 2 can be automatically extended to the case $r = 0$ as well.*

---

**Algorithm 1** The Splitting Stochastic Meta-algorithm

---

**Input** $\lambda > 0$, $r \in \{0\} \cup [n-1]$ (splitting parameter), $0 < \gamma \le 1/2$ (step size)

**Initialization** $h^{(0)} \in \mathbb{R}^d$

**For** $t = 1$ to $T$

    Receive   $Z_n^{(t)} = \left( X_n^{(t)}, \mathbf{y}_n^{(t)} \right)$

    Split   $Z_n^{(t)} = \left( Z_r^{(t)}, Z_{n-r}^{(t)} \right)$  with $\begin{cases} Z_r^{(t)} = \left( X_r^{(t)}, \mathbf{y}_r^{(t)} \right) \\ Z_{n-r}^{(t)} = \left( X_{n-r}^{(t)}, \mathbf{y}_{n-r}^{(t)} \right) \end{cases}$   $\longrightarrow$ Splitting step

    Build   $\bar{X}_r^{(t)}$ and $\bar{\mathbf{y}}_r^{(t)}$ by Eq. (11)

    Compute   $\nabla\mathcal{L}_r(h^{(t-1)}, \bar{X}_r^{(t)}, \bar{\mathbf{y}}_r^{(t)}) = \bar{X}_r^{(t)\top} \left( \bar{X}_r^{(t)} h^{(t-1)} - \bar{\mathbf{y}}_r^{(t)} \right)$

    Update   $h^{(t)} = h^{(t-1)} - \gamma\nabla\mathcal{L}_r(h^{(t-1)}, \bar{X}_r^{(t)}, \bar{\mathbf{y}}_r^{(t)})$   $\longrightarrow$ Stochastic step

**Return** $\bar{h}_{T,r,\lambda} = \frac{1}{T+1}\sum_{t=0}^{T} h^{(t)}$

---

### 3.2 The Stochastic Step

Once we have received a dataset and we have splitted it in order to compute the transfer risk $\mathcal{E}_r$ in Eq. (10), we apply Stochastic Gradient Descent (SGD) [13, 28] to minimize the function $\mathcal{E}_r$, see Alg. 1. We remark that, in our case, the application of the algorithm is slightly different to the classical setting of LS regression, since, at each iteration, we process a data point of the form $(X, \mathbf{y})$ with $X$ matrix and $\mathbf{y}$ vector, while, in the standard setting, usually we sample a vector $x$ and a real number $y$. More precisely, thanks to Prop. 2, introducing the function

$$\mathcal{L}_r(h, \bar{X}_r, \bar{\mathbf{y}}_r) = \frac{1}{2}\left\| \bar{X}_r h - \bar{\mathbf{y}}_r \right\|^2,$$

we can rewrite the transfer risk $\mathcal{E}_r$ in Eq. (10) as $\mathcal{E}_r(h) = \mathbb{E}_{\bar{X}_r, \bar{\mathbf{y}}_r}\mathcal{L}_r(h, \bar{X}_r, \bar{\mathbf{y}}_r)$. The estimator $\bar{h}_{T,r,\lambda}$ returned by Alg. 1 is given by the average of the iterations. The next section is devoted to the analysis of the statistical properties of this estimator, remembering that, since we are interested in testing the performance of the algorithm using all the $n$ data points, our final aim is to minimize $\mathcal{E}_n$ and not $\mathcal{E}_r$. Note, however, that the meta-algorithm allows us to minimize directly $\mathcal{E}_r$ for every $r \le n-1$, which may be useful when we observe future datasets of sample size smaller than $n$.

## 4 Statistical Analysis

In the proposition below, we give an upper bound on the expected *excess transfer risk* $\mathcal{E}_n(\bar{h}_{T,r,\lambda}) - \mathcal{E}_n(h_n^*)$ for Ex. 1. The bound suggests also how to choose the splitting parameter $r$ for that specific environment. A complete statistical analysis for more general environments remains an interesting open problem, which will be addressed in future work.

**Proposition 3.** *Assume $\mathcal{X} \subseteq \mathcal{B}_1$ and, for any $r \in \{0\} \cup [n-1]$ and $\lambda > 0$, let $\bar{h}_{T,r,\lambda}$ be the output of Alg. 1. Then, the expected excess transfer risk of the algorithm in Eqs. (8)-(9) with parameter $\bar{h}_{T,r,\lambda}$ trained with $n$ points over the environment in Ex. 1 is bounded by*

$$\mathbb{E}\big[\mathcal{E}_n(\bar{h}_{T,r,\lambda}) - \mathcal{E}_n(h_n^*)\big] \le \frac{(r/n + \lambda)^2}{\lambda^2}\frac{4K_\rho}{T+1}\Big(\frac{1}{\gamma}\big\|h^{(0)} - \bar{w}\big\|^2 + \Big(\sigma_w^2 + \Big(\frac{1}{n-r} + \frac{r}{(n\lambda)^2}\Big)\sigma_\epsilon^2\Big)d\Big),$$

*where the expectation is over the datasets $Z_n^{(1)}, \ldots, Z_n^{(T)}$ and $K_\rho$ is condition number of the environment defined as*

$$K_\rho = \frac{\mathbb{E}_{\mu\sim\rho}\big\|\Sigma_\mu\big\|_\infty}{\mathbb{E}_{\mu\sim\rho}\lambda_{\min}\big(\Sigma_\mu\big)}. \tag{13}$$

We now briefly describe the sketch of the proof of Prop. 3, all the details are reported in App. E.

**Proof.** We start observing that, since we are dealing with a LS problem (see Prop. 1), the excess transfer risk coincides with $\mathcal{E}_n(\bar{h}_{T,r,\lambda}) - \mathcal{E}_n(h_n^*) = \frac{1}{2}\big\|\bar{\Sigma}_n^{1/2}\big(\bar{h}_{T,r,\lambda} - h_n^*\big)\big\|^2$. Adding and subtracting

$\bar{\Sigma}_n^{1/2} h_r^*$ inside the norm, taking the expectation and using the inequality $\|a + b\|^2 \le 2\|a\|^2 + 2\|b\|^2$ for any two vectors $a, b \in \mathbb{R}^d$, we get that

$$\mathbb{E}\big[\mathcal{E}_n(\bar{h}_{T,r,\lambda}) - \mathcal{E}_n(h_n^*)\big] \le \underbrace{\mathbb{E}\Big[\big\|\bar{\Sigma}_n^{1/2}(\bar{h}_{T,r,\lambda} - h_r^*)\big\|^2\Big]}_{A} + \underbrace{\mathbb{E}\Big[\big\|\bar{\Sigma}_n^{1/2}(h_r^* - h_n^*)\big\|^2\Big]}_{B}. \qquad (14)$$

Thanks to the structure of the environment we are considering, we have that $h_r^* = \bar{w}$ for any $r$ (see Ex. 1), and consequently, the term $B$ vanishes. Regarding the term $A$, it is possible to show that

$$A \le \frac{2K_\rho(r/n + \lambda)^2}{\lambda^2} \mathbb{E}\big[\mathcal{E}_r(\bar{h}_{T,r,\lambda}) - \mathcal{E}_r(h_r^*)\big]$$

where $K_\rho$ is defined in Eq. (13). Finally, the term $\mathbb{E}\big[\mathcal{E}_r(\bar{h}_{T,r,\lambda}) - \mathcal{E}_r(h_r^*)\big]$ can be bounded via standard convergence rates for SGD for LS problem (see Thm. 12 in App. C). The result follows by estimating the quantities in this rate in the specific setting of Ex. 1 (see Thm. 20 in App. D). ∎

The bound in Prop. 3 is increasing in $r$, consequently, it suggests that, in the specific case of Ex. 1, the best choice of the optimal splitting parameter is $r = 0$, corresponding to using all the points in each dataset to test the inner algorithm that outputs $h$, for any training set. Our analysis reveals that this fact is due to both the specific characteristics of Ex. 1 and the normalization of the inner algorithm. Moreover, as we will see in the next section, this optimal choice of $r$ is confirmed also by our experiments. Finally, we observe that, differently from standard bounds for LTL [12, 20, 22], the bound in Prop. 3 is consistent as $T \to \infty$, for any fixed values of sample size $n$ and any fixed value of the hyper-parameter $\lambda$. We note that to get such strong guarantees for more general environments is more challenging since in general $h_r^* \ne h_n^*$ (see e.g. Ex. 2). Thus, as explained in the proof sketch above, the decomposition of the excess transfer risk usually involves a further term measuring the weighted distance between the solutions of the two different LS problems.

## 5 Experiments

We report the empirical evaluation of our LTL estimator on synthetic and real data[4]. In the experiments, $\lambda$ and the splitting parameter $r$ were tuned by cross-validation (see App. F for more details). Specifically, we considered 20 candidate values of $\lambda$ in the range $[10^{-6}, 10^2]$, logarithmic spacing.

**Synthetic Datasets.** We considered two different data generation protocols aimed at investigating empirically our analysis in Ex. 1 and Ex. 2. In both settings we considered a LTL scenario with 100 training datasets (tasks) observed one at the time. We used 50 tasks to perform model selection and 200 tasks for test. Each task corresponds to a dataset $(x_i, y_i)_{i=1}^n$, $x_i \in \mathbb{R}^d$ with $d = 30$ and $y_i$ generated according to the specific setting (see below). For the training tasks we fixed $n = 20$.

• **Ex. 1.** We generated the datasets according to an environment satisfying the hypotheses of Ex. 1. For each task we generated the inputs $x$ uniformly on the sphere of radius 1, the task vector $w$ from a Gaussian distribution with mean $\bar{w} = \mathbf{4} \in \mathbb{R}^d$, the vector with entries all equal to 4, and standard deviation $\sigma_w = 1$. The labels were generated as $y = \langle x, w \rangle + \epsilon$ with $\epsilon$ sampled from a zero-mean Gaussian distribution, with standard deviation chosen to have signal-to-noise ratio equal to 5 for each task. Fig. 1 (Left) reports the performance of our LTL estimator for different split sizes $r$. Interestingly, we observe exactly the behavior predicted by the theoretical analysis for Ex. 1: in this setting, the optimal LTL strategy is to set $r = 0$. This motivates us to consider the following more challenging environment.

• **Ex. 2.** We generated the data according to the environment described in Ex. 2 for $K = 2$ groups of tasks. The two marginals $p_k$ were chosen as Gaussian distributions with same standard deviation $\sigma = 1$ and means $m_1 = \mathbf{0}$ and $m_2 = \mathbf{2}$. Analogously, the task vectors $w$ were sampled from one of two Gaussians with same standard deviation $\sigma = 1$ and means respectively $\bar{w}_1 = \mathbf{2}$ and $\bar{w}_2 = \mathbf{4}$. The two clusters were randomly selected with same probability $\nu_k = 1/2$, $k \in [2]$. We generated $y = \langle x, w \rangle + \epsilon$ with $\epsilon$ sampled with the same strategy as above. Fig. 1 (Right) reports the performance of our LTL estimator for different split sizes $r$. Here, differently from the previous case, the best choice for $r$ is a trade-off between 0 and $n - 1$. We also compared our LTL estimator with the ITL baseline (applying Ridge Regression with $h = 0$ independently for each task) as the number of tasks

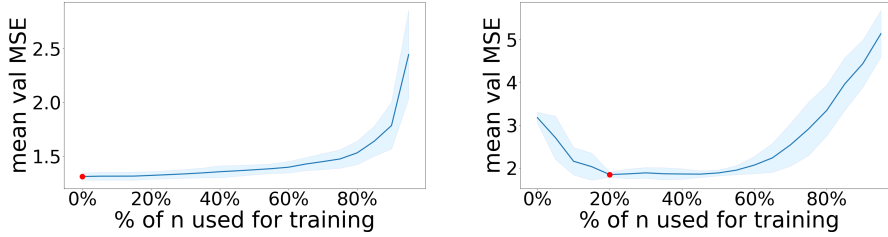

Figure 1: Mean square error (MSE) on the validation tasks after 100 training tasks of the LTL estimator for varying dataset split sizes. Data generated according to (Left) Ex. 1 and (Right) Ex. 2. The dot indicates the minimum. The results are averaged over 30 independent runs (dataset generations).

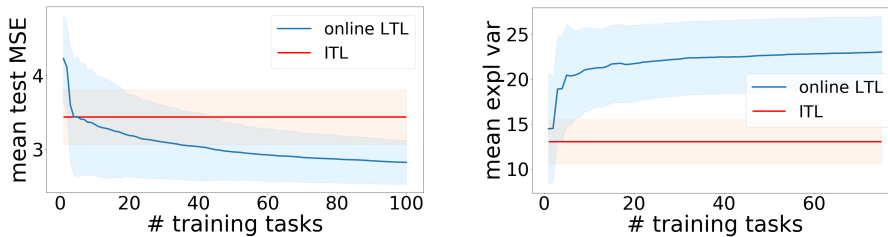

Figure 2: Performance of LTL vs ITL with respect to an increasing number of tasks. (Left) MSE on the test tasks for the dataset generation of Ex. 2. (Right) Explained variance on the test tasks for the School Dataset. The results are averaged over 30 independent runs (datasets/train-split generations).

increases. This comparison is reported in Fig. 2 (Left). For LTL we report the performance for the best split sizes $r$ chosen by cross-validation. Note that LTL rapidly improves with respect to ITL.

**School Dataset.** We compared the performance of LTL and ITL on the School dataset (see [2]), which contains 139 tasks of dimension $d = 26$ each. For all the experiments we randomly sampled 75 tasks for training, 25 for validation and the rest for test. Performance of the two methods was measured in terms of the *explained variance* [2] (the higher the better). Fig. 2 (Right) reports the performance of the two methods. Also here, we report the performance of LTL for the best $r$ chosen by cross-validation. We observe that, again, LTL very rapidly outperforms ITL. This experiment shows that the LTL framework around a common mean is a transfer-knowledge method that can be successfully applied, not only to synthetic toy datasets, but also to appropriate real data.

# 6 Conclusion and Future Work

In this paper we considered the Learning-To-Learn setting in which the underlying algorithm is Ridge Regression around a common mean. We showed that the associated LTL problem coincides with a Least Squares problem with data distribution induced by the tasks' meta-distribution and, in order to solve it, we presented a novel stochastic meta-algorithm based on direct optimization of the transfer risk. Under specific assumptions of the environment, we derived an analysis of the generalization performance of the method, which highlights the role of the splitting parameter in the learning process. Preliminary experiments confirmed our theoretical findings. Future work will be devoted to extend the statistical analysis of our meta-algorithm to more general environments and to compare our approach with the more standard one working with the future empirical risk.

**Acknowledgments**

This work was supported in part by the UK Defence Science and Technology Laboratory (Dstl) and Engineering and Physical Research Council (EPSRC) under grant EP/P009069/1. This is part of the collaboration between US DOD, UK MOD and UK EPSRC under the Multidisciplinary University Research Initiative.

## Footnotes

[1]Specifically, $\rho$ is a distribution on the set of probability distributions on $\mathcal{Z}$ and each dataset is observed by first sampling a task $\mu \sim \rho$ and then sampling a dataset $Z_n \sim \mu^n$ of $n$ i.i.d. points. However, with some abuse of notation, we refer to the environment both as the meta-distribution and the induced distribution on $n$-samples.

[2]For simplicity we assume that every dataset is composed by the same number of points $n$.

[3]This is the meta-version of a frequent assumption in single-task LS literature, where usually the invertibility of the covariance matrix is required, see e.g. [13] and references therein.

[4]The code used for the following experiments is available at *https://github.com/dstamos*

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
