[Supplementary Material]

## Appendix

In this appendix, we report proofs of the results presented in the paper and also additional remarks. We start giving some auxiliary results in App. A and the proofs of the statements made in Sec. 2 and Sec. 3 in App. B. In App. C we present the convergence rate of SGD for a general vector-valued LS problem (also known as Least Mean Squares (LMS) algorithm). This rate is then specialized in App. D to our Alg. 1 minimizing the LS function $\mathcal{E}_r$. App. E contains the proof of the main result in the paper (Prop. 3) and, finally, in App. F, we describe how to tune the hyper-parameters in the LTL setting described in the paper by a cross-validation procedure. In the sequel we will use the notation already introduced in the paper.

## A  Auxiliary Results

In this section, we give some technical tools that will be used in the sequel of this appendix. We denote by $\mathbb{S}^d$, $\mathbb{S}^d_{++}$ and $\mathbb{S}^d_+$ the sets of the real $d \times d$ symmetric, symmetric positive definite and symmetric positive semidefinite matrices, respectively. We refer to the book [5] for more details.

**Lemma 4.** *Let $U \in \mathbb{S}^d_+$ and let $V \in \mathbb{R}^{d \times m}$, then $V^\top U V \in \mathbb{S}^m_+$.*

**Proof.** For any $a \in \mathbb{R}^m$, defining $b = Va \in \mathbb{R}^d$ and using the assumption $U \in \mathbb{S}^d_+$, we have that $a^\top V^\top U V a = b^\top U b \geq 0$. This proves the desired statement. ∎

A direct consequence of the above remark is the following fact.

**Corollary 5.** *Let $W \in \mathbb{S}^d_+$, then $W^2 \preceq \|W\|_\infty W$.*

**Proof.** The statement directly follows from applying Lemma 4 with $V = W^{1/2}$ and $U = \|W\|_\infty I - W$ and using the fact $W \preceq \|W\|_\infty I$. Specifically, $W^2 = W^{1/2} W W^{1/2} \preceq \|W\|_\infty W$. ∎

**Lemma 6.** *Let $A \in \mathbb{S}^d_{++}$ and $B \in \mathbb{S}^d_+$ such that $AB = BA$, then $AB \in \mathbb{S}^d_+$.*

**Proof.** We first note that $AB$ is symmetric. Indeed, we have that $(AB)^\top = B^\top A^\top = BA = AB$. Now, we observe that $AB$ is similar to $A^{1/2}BA^{1/2}$, specifically, we have

$$AB = A^{1/2}A^{-1/2}ABA^{1/2}A^{-1/2} \sim A^{-1/2}ABA^{1/2} = A^{1/2}BA^{1/2}.$$

Consequently, the eigenvalues of $AB$ are the same of $A^{1/2}BA^{1/2}$ and the statement follows from Lemma 4, according to which $A^{1/2}BA^{1/2} \in \mathbb{S}^d_+$. ∎

**Lemma 7.** *The function $\lambda_{\min} : \mathbb{S}^d \to \mathbb{R}$ is matrix-concave.*

**Proof.** We recall that for any $A \in \mathbb{S}^d$, we can write

$$\lambda_{\min}(A) = \inf_{v \in \mathbb{R}^d \setminus \{\mathbf{0}\}} \frac{v^\top A v}{v^\top v} = \inf_{v \in \mathbb{R}^d \setminus \{\mathbf{0}\}} f_v(A),$$

where, for any $v \in \mathbb{R}^d \setminus \{\mathbf{0}\}$, we have introduced the linear functions $f_v : \mathbb{S}^d \to \mathbb{R}$, $A \in \mathbb{S}^d \mapsto \frac{v^\top A v}{v^\top v} \in \mathbb{R}$. Hence, the concavity of $\lambda_{\min}(\cdot)$ derives from the fact that the infimum of linear functions is a concave function. ∎

**Lemma 8.** *Let $A \in \mathbb{S}^d_{++}$ and $B \in \mathbb{S}^d_+$. Then, $ABA \in \mathbb{S}^d_+$ and the following lower bound on the smallest eigenvalue of $ABA$ holds*

$$\lambda_{\min}(ABA) \geq \lambda_{\min}(B)\lambda_{\min}(A)^2.$$

**Proof.** We start observing that $ABA \in \mathbb{S}_+^d$, thanks to Lemma 4. Moreover, as already observed in the above lemma, we have that

$$
\begin{aligned}
\lambda_{\min}(ABA) &= \inf_{v \in \mathbb{R}^d \setminus \{\mathbf{0}\}} \frac{v^\top ABA v}{v^\top v} = \inf_{v \in \mathbb{R}^d \setminus \{\mathbf{0}\}} \frac{v^\top ABA v}{v^\top A^2 v} \frac{v^\top A^2 v}{v^\top v} \\
&\geq \inf_{v \in \mathbb{R}^d \setminus \{\mathbf{0}\}} \frac{v^\top ABA v}{v^\top A^2 v} \inf_{w \in \mathbb{R}^d \setminus \{\mathbf{0}\}} \frac{w^\top A^2 w}{w^\top w} \\
&= \inf_{u \in \mathbb{R}^d \setminus \{\mathbf{0}\}} \frac{u^\top B u}{u^\top u} \inf_{w \in \mathbb{R}^d \setminus \{\mathbf{0}\}} \frac{w^\top A^2 w}{w^\top w} \\
&= \lambda_{\min}(B) \lambda_{\min}(A^2),
\end{aligned}
$$

where the second and the third equality hold thanks to the fact that $A \in \mathbb{S}_{++}^d$, hence $A^2 \in \mathbb{S}_{++}^d$, and consequently, for any $v \in \mathbb{R}^d \setminus \{\mathbf{0}\}$, $Av \neq \mathbf{0}$. The inequality above holds since, for any two non-negative functions $g$ and $f$, we have that $\inf_v g(v) f(v) \geq \inf_v g(v) \inf_w f(w)$; in our case, the two functions are non-negative since $A^2 \in \mathbb{S}_{++}^d$ and, as already observed, $ABA \in \mathbb{S}_+^d$. The statement directly follows observing that $\lambda_{\min}(A^2) = \lambda_{\min}(A)^2$. ∎

The following result is an extension of the Neumann series to matrices.

**Lemma 9** (Exercise I.2.6 in [5])**.** *Let $A \in \mathbb{S}^d$ such that $0 \prec A \prec I$, then*

$$
\sum_{k=1}^{\infty} (I - A)^k = A^{-1}(I - A).
$$

We conclude this section with a technical lemma which will be used in the following.

**Lemma 10.** *For the indexes $t, j \in \{i_{\min}, \ldots, i_{\max}\}$ consider the sequence of numbers $(a_{t,j})_{t,j}$, where, for any $t, j \in \{i_{\min}, \ldots, i_{\max}\}$, $a_{t,j} \in \mathbb{R}$ and $a_{t,j} = a_{j,t}$. Then, we can write*

$$
\sum_{t=i_{\min}}^{i_{\max}} \sum_{j=i_{\min}}^{i_{\max}} a_{t,j} = \sum_{t=i_{\min}}^{i_{\max}} a_{t,t} + 2 \sum_{t=i_{\min}}^{i_{\max}-1} \sum_{j=t+1}^{i_{\max}} a_{t,j}.
$$

**Proof.** We can interpret each element $a_{t,j}$ in the sequence as the entry $A_{t,j}$ of a symmetric matrix A with rows and columns indexes in the set $\{i_{\min}, \ldots, i_{\max}\}$. The statement follows by first summing the elements in the diagonal and then the elements of the upper and lower triangular parts, observing that, thanks to the symmetry of the matrix, these two last parts bring exactly the same contribution in the sum. Specifically, we have that

$$
\sum_{t=i_{\min}}^{i_{\max}} \sum_{j=i_{\min}}^{i_{\max}} a_{t,j} = \sum_{t=i_{\min}}^{i_{\max}} a_{t,t} + \sum_{t=i_{\min}}^{i_{\max}} \sum_{j \neq t} a_{t,j},
$$

where, thanks to the symmetry,

$$
\sum_{t=i_{\min}}^{i_{\max}} \sum_{j \neq t} a_{t,j} = 2 \sum_{t=i_{\min}}^{i_{\max}-1} \sum_{j=t+1}^{i_{\max}} a_{t,j}.
$$

∎

# B   Proofs of Results in Sec. 2 and Sec. 3

In order to prove both Prop. 1 and Prop. 2, we will need the following result, which will be used also in the proof of Prop. 3 in App. E.

**Lemma 11** (A Lower Bound on $\lambda_{\min}(\bar{\Sigma}_r)$ and an Upper Bound on $\left\lVert \bar{\Sigma}_r \right\rVert_\infty$)**.** *Let $\mathcal{X} \subseteq \mathcal{B}_1$ and let, for any $r \in \{0\} \cup [n]$, $\bar{\Sigma}_r$ be defined as in Prop. 1 and Prop. 2. Then, for any $r \in \{0\} \cup [n]$, the following inequalities hold.*

$$
\frac{\lambda^2 \mathbb{E}_{\mu \sim \rho} \lambda_{\min}(\Sigma_\mu)}{(r/n + \lambda)^2} \leq \lambda_{\min}(\bar{\Sigma}_r) \leq \left\lVert \bar{\Sigma}_r \right\rVert_\infty \leq \mathbb{E}_{\mu \sim \rho} \left\lVert \Sigma_\mu \right\rVert_\infty.
$$

**Proof.** We recall by definition that, for any $r \in [n]$,

$$\bar{\Sigma}_r = \lambda^2 \mathbb{E}\left[C_{\lambda,r}^{-1} \Sigma_\mu C_{\lambda,r}^{-1}\right].$$

We start from proving the lower bound on $\lambda_{\min}(\bar{\Sigma}_r)$. As already observed in Lemma 7, the function $\lambda_{\min} : \mathbb{S}^d \to \mathbb{R}$ is matrix-concave, hence, by Jensen's inequality, for any $M \in \mathbb{S}^d$, we have that

$$\lambda_{\min}\big(\mathbb{E}\big[M\big]\big) \geq \mathbb{E}\big[\lambda_{\min}(M)\big].$$

Consequently, using the facts $\|X_r\|_\infty^2 \leq \|X_r\|^2 \leq r$ (since $\mathcal{X} \subseteq \mathcal{B}_1$) and $\lambda_{\min}\big(C_{\lambda,r}^{-1}\big) = \|C_{\lambda,r}\|_\infty^{-1}$ (since $C_{\lambda,r}^{-1} \in \mathbb{S}_{++}^d$) and applying Lemma 8 to the matrices $A = C_{\lambda,r}^{-1} \in \mathbb{S}_{++}^d$ and $B = \Sigma_\mu \in \mathbb{S}_+^d$, we obtain that

$$
\begin{aligned}
\lambda_{\min}\left(\mathbb{E}\left[C_{\lambda,r}^{-1} \Sigma_\mu C_{\lambda,r}^{-1}\right]\right) &\geq \mathbb{E}\left[\lambda_{\min}\left(C_{\lambda,r}^{-1} \Sigma_\mu C_{\lambda,r}^{-1}\right)\right] \geq \mathbb{E}\left[\lambda_{\min}\left(\Sigma_\mu\right)\lambda_{\min}\left(C_{\lambda,r}^{-1}\right)^2\right] \\
&= \mathbb{E}\left[\lambda_{\min}\left(\Sigma_\mu\right)\left(\|C_{\lambda,r}\|_\infty\right)^{-2}\right] = \mathbb{E}\left[\lambda_{\min}\left(\Sigma_\mu\right)\left(\|X_r\|_\infty^2/n + \lambda\right)^{-2}\right] \\
&\geq \frac{\mathbb{E}_{\mu \sim \rho}\lambda_{\min}\left(\Sigma_\mu\right)}{\left(r/n + \lambda\right)^2}.
\end{aligned}
$$

Multiplying by $\lambda^2$, the result follows. As regards the upper bound on $\|\bar{\Sigma}_r\|_\infty$, we proceed as follows. Exploiting the fact $\|C_{\lambda,r}^{-1}\|_\infty \leq 1/\lambda$, we can write

$$\|\bar{\Sigma}_r\|_\infty \leq \lambda^2 \mathbb{E}\left[\|C_{\lambda,r}^{-1} \Sigma_\mu C_{\lambda,r}^{-1}\|_\infty\right] \leq \lambda^2 \mathbb{E}\left[\|C_{\lambda,r}^{-1}\|_\infty \|\Sigma_\mu\|_\infty \|C_{\lambda,r}^{-1}\|_\infty\right] \leq \mathbb{E}_{\mu \sim \rho}\|\Sigma_\mu\|_\infty. \quad (15)$$

We observe that the previous steps hold also for the extreme case $r = 0$, where $C_{\lambda,0} = \lambda I$ and $\bar{\Sigma}_0 = \mathbb{E}_{\mu \sim \rho}\Sigma_\mu$. However, in such a case, we can also directly get the statement in the proposition by simply applying Jensen's inequality, getting both $\lambda_{\min}\big(\mathbb{E}_{\mu \sim \rho}\Sigma_\mu\big) \geq \mathbb{E}_{\mu \sim \rho}\lambda_{\min}\big(\Sigma_\mu\big)$ and $\|\mathbb{E}_{\mu \sim \rho}\Sigma_\mu\|_\infty \leq \mathbb{E}_{\mu \sim \rho}\|\Sigma_\mu\|_\infty$. ∎

We now are ready to prove Prop. 1.

**Proposition 1** (LS Problem Around a Common Mean for $\mathcal{E}_n$)**.** *For any $\lambda > 0$ and $h \in \mathbb{R}^d$, the transfer risk $\mathcal{E}_n$ in Eq. (1) of the learning algorithm $w_h$ in Eqs. (3)-(4), can be rewritten as*

$$\mathcal{E}_n(h) = \frac{1}{2}\mathbb{E}_{\bar{x}_n, \bar{y}_n}\left[\left(\langle \bar{x}_n, h \rangle - \bar{y}_n\right)^2\right] \quad (5)$$

*where the meta-data are given by*

$$\bar{x}_n = \lambda C_{\lambda,n}^{-1} x, \quad \text{and } \bar{y}_n = y - \left\langle \bar{x}_n, \frac{X_n^\top \mathbf{y}_n}{\lambda n} \right\rangle.$$

*Moreover, under Asm. 1, the meta-covariance matrix $\bar{\Sigma}_n = \mathbb{E}\bar{x}_n \bar{x}_n^\top = \lambda^2 \mathbb{E}\big[C_{\lambda,n}^{-1} \Sigma_\mu C_{\lambda,n}^{-1}\big]$ is invertible and $h_n^* = \bar{\Sigma}_n^{-1}\mathbb{E}[\bar{y}_n \bar{x}_n]$ is the unique minimizer of the LS function in Eq. (5). In such a case, letting $v = w - h_n^*$, we have that $\bar{y}_n = \langle \bar{x}_n, h_n^* \rangle + \bar{\epsilon}_n$, with $\bar{\epsilon}_n = \epsilon + \left\langle \bar{x}_n, v - \frac{X_n^\top \epsilon_n}{\lambda n} \right\rangle$.*

**Proof.** The rewriting of the transfer risk in Eq. (5) is a direct consequence of the closed form of the algorithm in Eq. (4). The invertibility of the meta-covariance matrix is a direct consequence of Lemma 11 for $r = n$ and the requirement $\mathbb{E}_{\mu \sim \rho}\lambda_{\min}\big(\Sigma_\mu\big) > 0$ in Asm. 1. This implies that the LS function is strictly convex and, consequently, it admits a unique minimizer $h_n^*$. The closed form of this minimizer directly follows from the optimality conditions (normal equations) of the problem. Now, thanks to Asm. 1, using the linear model equations

$$y = \langle x, w \rangle + \epsilon \qquad \mathbf{y}_n = X_n w + \epsilon_n$$

and letting $w = h_n^* + v$, for some $v \in \mathbb{R}^d$, we can rewrite

$$
\begin{aligned}
\bar{y}_n &= y - \left\langle \bar{x}_n, \frac{X_n^\top \mathbf{y}_n}{\lambda n} \right\rangle = y - \left\langle x, C_{\lambda,n}^{-1} \frac{X_n^\top \mathbf{y}_n}{n} \right\rangle \\
&= \langle x, w \rangle + \epsilon - \left\langle x, C_{\lambda,n}^{-1} \frac{X_n^\top (X_n w + \boldsymbol{\epsilon}_n)}{n} \right\rangle \\
&= \left\langle x, C_{\lambda,n}^{-1} \Big( C_{\lambda,n} - \frac{X_n^\top X_n}{n} \Big) w \right\rangle + \epsilon - \left\langle x, C_{\lambda,n}^{-1} \frac{X_n^\top \boldsymbol{\epsilon}_n}{n} \right\rangle \\
&= \left\langle x, \lambda C_{\lambda,n}^{-1} w \right\rangle + \epsilon - \left\langle x, C_{\lambda,n}^{-1} \frac{X_n^\top \boldsymbol{\epsilon}_n}{n} \right\rangle \\
&= \langle \bar{x}_n, w \rangle + \epsilon - \left\langle \bar{x}_n, \frac{X_n^\top \boldsymbol{\epsilon}_n}{\lambda n} \right\rangle \\
&= \langle \bar{x}_n, h_n^* \rangle + \epsilon + \left\langle \bar{x}_n, v - \frac{X_n^\top \boldsymbol{\epsilon}_n}{\lambda n} \right\rangle.
\end{aligned}
\tag{16}
$$

We conclude that the noise on the meta-labels is heteroscedastic, since it is given by

$$
\bar{\epsilon}_n = \bar{y}_n - \langle \bar{x}_n, h_n^* \rangle = \epsilon + \left\langle \bar{x}_n, v - \frac{X_n^\top \boldsymbol{\epsilon}_n}{\lambda n} \right\rangle.
$$

∎

We now proceed with the proof of Ex. 1 and the remarks associated to it. Before doing this, we point out the following aspect which will be used throughout the appendix.

**Remark 8.** *According to the data-generation procedure described in Asm. 1, the samplings of the quantities $x, w, \epsilon$ are independent one each other, conditioning with respect to the marginal distribution $p$.*

**Example 1.** *Let $\mathcal{X} \subseteq \mathcal{B}_1$ and let the environment $\rho$ satisfy Asm. 1. Furthermore, assume for almost every $p$ that: i) $\eta \mid p$ has variance bounded by $\sigma_\epsilon^2$, for $\sigma_\epsilon \geq 0$, ii) $\mathbb{E}[w|p] = \bar{w}$, and iii) $\mathbb{E}\big[(w - \bar{w})(w - \bar{w})^\top | p\big] \preceq \sigma_w^2 I$, for $\sigma_w \geq 0$. Then, for any $\lambda > 0$ and $h \in \mathbb{R}^d$, $h_n^* = \bar{w}$ and, letting $A_n = C_{\lambda,n}^{-1} x x^\top C_{\lambda,n}^{-1}$, we have*

$$
\mathcal{E}_n(h) \leq \frac{1}{2} \big\| \bar{\Sigma}_n^{1/2} (h - \bar{w}) \big\|^2 + \frac{\sigma_w^2}{2} \mathrm{tr}(\bar{\Sigma}_n) + \frac{\sigma_\epsilon^2}{2} \Big( 1 + \mathrm{tr}\Big( \mathbb{E}\Big[ \frac{X_n^\top X_n A_n}{n^2} \Big] \Big) \Big).
\tag{6}
$$

**Proof.** In the following, because of readability, we condense all the expectations (the one according the sampling of the task $\mu = (w, p, \eta) \sim \rho$, the one referring to the sampling of the training dataset $Z_n \sim \mu^n$ and the one related to the sampling of the test point $z \sim \mu$) in only one symbol. In this way, the transfer risk $\mathcal{E}_n$ of the algorithm $w_h$ on any environment satisfying Asm. 1 can be rewritten as

$$
\begin{aligned}
\mathcal{E}_n(h) &= \frac{1}{2} \mathbb{E}\Big[ \big( \langle x, w_h(Z_n) \rangle - y \big)^2 \Big] \\
&= \frac{1}{2} \mathbb{E}\Big[ \big( \langle x, w_h(Z_n) \rangle - \langle x, w \rangle - \epsilon \big)^2 \Big] \\
&= \frac{1}{2} \mathbb{E}\Big[ (w_h(Z_n) - w)^\top x x^\top (w_h(Z_n) - w) \Big] + \frac{\mathbb{E}\big[\epsilon^2\big]}{2},
\end{aligned}
$$

where in the second equality we have used the linear model equation $y = \langle x, w \rangle + \epsilon$ and in the third equality we have exploited the fact that the noise is zero-mean, more precisely, thanks to Rem. 8,

$$
\mathbb{E}\Big[ \epsilon \langle x, w_h(Z_n) - w \rangle \Big] = \mathbb{E}\Big[ \mathbb{E}\big[ \epsilon \langle x, w_h(Z_n) - w \rangle | p \big] \Big] = \mathbb{E}\Big[ \mathbb{E}\big[\epsilon|p\big] \mathbb{E}\big[ \langle x, w_h(Z_n) - w \rangle | p \big] \Big] = 0.
$$

Using the closed form of the algorithm in Eq. (4) and the equation $\mathbf{y}_n = X_n w + \boldsymbol{\epsilon}_n$ deriving from Asm. 1, a direct computation gives that

$$
w_h(Z_n) - w = \lambda C_{\lambda,n}^{-1}(h - w) + C_{\lambda,n}^{-1} \frac{X_n^\top \boldsymbol{\epsilon}_n}{n}.
$$

Consequently, we can write

$$
\begin{aligned}
(w_h(Z_n) - w)^\top x x^\top (w_h(Z_n) - w) = {} & \lambda^2 (h - w)^\top C_{\lambda,n}^{-1} x x^\top C_{\lambda,n}^{-1} (h - w) \\
& + \frac{\epsilon_n^\top X_n}{n} C_{\lambda,n}^{-1} x x^\top C_{\lambda,n}^{-1} \frac{X_n^\top \epsilon_n}{n} \\
& + 2\lambda (h - w)^\top C_{\lambda,n}^{-1} x x^\top C_{\lambda,n}^{-1} \frac{X_n^\top \epsilon_n}{n}.
\end{aligned} \tag{17}
$$

Hence, recalling the definition of the matrix $A_n = C_{\lambda,n}^{-1} x x^\top C_{\lambda,n}^{-1}$, we have that

$$
\begin{aligned}
(w_h(Z_n) - w)^\top x x^\top (w_h(Z_n) - w) = {} & \lambda^2 (h - w)^\top A_n (h - w) + \frac{\epsilon_n^\top X_n A_n X_n^\top \epsilon_n}{n^2} \\
& + 2\lambda (h - w)^\top \frac{A_n X_n^\top \epsilon_n}{n}.
\end{aligned} \tag{18}
$$

Consequently, taking the expectation of Eq. (18) with respect to the sampling of the task $\mu = (w, p, \eta) \sim \rho$ and with respect to the sampling of the data $Z_n \sim \mu^n$ and $z \sim \mu$, we obtain that

$$
\mathcal{E}_n(h) = \frac{1}{2} \mathbb{E}\left[ \lambda^2 (h - w)^\top A_n (h - w) + \frac{\epsilon_n^\top X_n A_n X_n^\top \epsilon_n}{n^2} \right] + \frac{\mathbb{E}[\epsilon^2]}{2}, \tag{19}
$$

where we have exploited again the fact that the noise distribution has zero-mean, more precisely, using Rem. 8, we have that

$$
\mathbb{E}\left[ (h - w)^\top \frac{A_n X_n^\top \epsilon_n}{n} \right] = \mathbb{E}\left[ \mathbb{E}\left[ (h - w)^\top \frac{A_n X_n^\top \epsilon_n}{n} \Big| p \right] \right] = \mathbb{E}\left[ \mathbb{E}\left[ (h - w)^\top \frac{A_n X_n^\top}{n} \Big| p \right] \mathbb{E}[\epsilon_n | p] \right] = 0.
$$

Hence, letting $w = \bar{w} + v$, with $v \in \mathbb{R}^d$, we can rewrite

$$
\mathcal{E}_n(h) = \frac{1}{2} \mathbb{E}\left[ \lambda^2 (h - \bar{w})^\top A_n (h - \bar{w}) - 2\lambda^2 (h - \bar{w})^\top A_n v + \lambda^2 v^\top A_n v + \frac{\epsilon_n^\top X_n A_n X_n^\top \epsilon_n}{n^2} \right] + \frac{\mathbb{E}[\epsilon^2]}{2}.
$$

We now observe that, thanks to condition $ii)$, $\mathbb{E}[v | p] = 0$ and, consequently, by Rem. 8, we have that

$$
\mathbb{E}\left[ (h - \bar{w})^\top A_n v \right] = (h - \bar{w})^\top \mathbb{E}\left[ \mathbb{E}[A_n v | p] \right] = (h - \bar{w})^\top \mathbb{E}\left[ \mathbb{E}[A_n | p] \mathbb{E}[v | p] \right] = 0.
$$

Hence, observing that

$$
\begin{aligned}
\lambda^2 v^\top A_n v &= \operatorname{tr}(v^\top \lambda^2 A_n v) = \operatorname{tr}(v v^\top \lambda^2 A_n) \\
\frac{\epsilon_n^\top X_n A_n X_n^\top \epsilon_n}{n^2} &= \operatorname{tr}\left( \epsilon_n \epsilon_n^\top \frac{X_n A_n X_n^\top}{n^2} \right),
\end{aligned} \tag{20}
$$

and exploiting the relation $\bar{\Sigma}_n = \mathbb{E}[\lambda^2 A_n]$, we can conclude that

$$
\mathcal{E}_n(h) = \frac{1}{2} (h - \bar{w})^\top \bar{\Sigma}_n (h - \bar{w}) + \frac{1}{2} \operatorname{tr}\left( \mathbb{E}[v v^\top \lambda^2 A_n] \right) + \frac{1}{2} \operatorname{tr}\left( \mathbb{E}\left[ \epsilon_n \epsilon_n^\top \frac{X_n A_n X_n^\top}{n^2} \right] \right) + \frac{\mathbb{E}[\epsilon^2]}{2}.
$$

From this last equation, taking the derivative with respect to $h$ and exploiting the fact that the covariance matrix $\bar{\Sigma}_n$ is invertible (see Prop. 1), we conclude that the unique minimizer of $\mathcal{E}_n(h)$ coincides with $h_n^* = \bar{w}$. The upper bound on $\mathcal{E}_n(h)$ given in the last statement of the example directly follows from the following steps. We start observing that, by Rem. 8, we can rewrite

$$
\mathbb{E}[v v^\top \lambda^2 A_n] = \mathbb{E}\left[ \mathbb{E}[v v^\top \lambda^2 A_n | p] \right] = \mathbb{E}\left[ \mathbb{E}[v v^\top | p] \mathbb{E}[\lambda^2 A_n | p] \right]
$$

and, consequently,

$$
\begin{aligned}
\operatorname{tr}\left( \mathbb{E}[v v^\top \lambda^2 A_n] \right) &= \mathbb{E}\left[ \operatorname{tr}\left( \mathbb{E}[v v^\top | p] \mathbb{E}[\lambda^2 A_n | p] \right) \right] \\
&= \mathbb{E}\left[ \operatorname{tr}\left( \mathbb{E}[\lambda^2 A_n | p]^{1/2} \mathbb{E}[v v^\top | p] \mathbb{E}[\lambda^2 A_n | p]^{1/2} \right) \right].
\end{aligned} \tag{21}
$$

Now, thanks to assumption $iii)$, we have that $\mathbb{E}[v v^\top | p] \preceq \sigma_w^2 I$, hence, applying twice Lemma 4 with

$$
U = \begin{cases} \mathbb{E}[v v^\top | p] \\ \sigma_w^2 I - \mathbb{E}[v v^\top | p] \end{cases} \qquad V = \mathbb{E}[\lambda^2 A_n | p]^{1/2},
$$

we have that

$$0 \preceq \mathbb{E}\big[\lambda^2 A_n | p\big]^{1/2} \mathbb{E}\big[v v^\top | p\big] \mathbb{E}\big[\lambda^2 A_n | p\big]^{1/2} \preceq \sigma_w^2 \mathbb{E}\big[\lambda^2 A_n | p\big].$$

Consequently, taking the trace of the above inequality, we can continue Eq. (21) as follows

$$\mathrm{tr}\big(\mathbb{E}\big[v v^\top \lambda^2 A_n\big]\big) \leq \sigma_w^2 \mathbb{E}\big[\mathrm{tr}\big(\mathbb{E}\big[\lambda^2 A_n | p\big]\big)\big] = \sigma_w^2 \mathrm{tr}(\bar{\Sigma}_n).$$

In a similar way, exploiting again Rem. 8, we observe that

$$\mathbb{E}\Big[\boldsymbol{\epsilon}_n \boldsymbol{\epsilon}_n^\top \frac{X_n A_n X_n^\top}{n^2}\Big] = \mathbb{E}\Big[\mathbb{E}\Big[\boldsymbol{\epsilon}_n \boldsymbol{\epsilon}_n^\top \frac{X_n A_n X_n^\top}{n^2}\Big| p\Big]\Big] = \mathbb{E}\Big[\mathbb{E}\big[\boldsymbol{\epsilon}_n \boldsymbol{\epsilon}_n^\top | p\big] \mathbb{E}\Big[\frac{X_n A_n X_n^\top}{n^2}\Big| p\Big]\Big]$$

and, consequently,

$$\begin{aligned}
\mathrm{tr}\Big(\mathbb{E}\Big[\boldsymbol{\epsilon}_n \boldsymbol{\epsilon}_n^\top \frac{X_n A_n X_n^\top}{n^2}\Big]\Big) &= \mathbb{E}\Big[\mathrm{tr}\Big(\mathbb{E}\big[\boldsymbol{\epsilon}_n \boldsymbol{\epsilon}_n^\top | p\big] \mathbb{E}\Big[\frac{X_n A_n X_n^\top}{n^2}\Big| p\Big]\Big)\Big] \\
&= \mathbb{E}\Big[\mathrm{tr}\Big(\mathbb{E}\Big[\frac{X_n A_n X_n^\top}{n^2}\Big| p\Big]^{1/2} \mathbb{E}\big[\boldsymbol{\epsilon}_n \boldsymbol{\epsilon}_n^\top | p\big] \mathbb{E}\Big[\frac{X_n A_n X_n^\top}{n^2}\Big| p\Big]^{1/2}\Big)\Big].
\end{aligned} \tag{22}$$

Now, thanks to assumption $i)$ and the independence of the points in the datasets, we have that $\mathbb{E}\big[\boldsymbol{\epsilon}_n \boldsymbol{\epsilon}_n^\top | p\big] \preceq \sigma_\epsilon^2 I$, hence, applying twice Lemma 4 with

$$U = \begin{cases} \mathbb{E}\big[\boldsymbol{\epsilon}_n \boldsymbol{\epsilon}_n^\top | p\big] \\ \sigma_\epsilon^2 I - \mathbb{E}\big[\boldsymbol{\epsilon}_n \boldsymbol{\epsilon}_n^\top | p\big] \end{cases} \qquad V = \mathbb{E}\Big[\frac{X_n A_n X_n^\top}{n^2}\Big| p\Big]^{1/2},$$

we get

$$0 \preceq \mathbb{E}\Big[\frac{X_n A_n X_n^\top}{n^2}\Big| p\Big]^{1/2} \mathbb{E}\big[\boldsymbol{\epsilon}_n \boldsymbol{\epsilon}_n^\top | p\big] \mathbb{E}\Big[\frac{X_n A_n X_n^\top}{n^2}\Big| p\Big]^{1/2} \preceq \sigma_\epsilon^2 \mathbb{E}\Big[\frac{X_n A_n X_n^\top}{n^2}\Big| p\Big].$$

Consequently, taking the trace of the above inequality, we can continue Eq. (22) as follows

$$\mathrm{tr}\Big(\mathbb{E}\Big[\boldsymbol{\epsilon}_n \boldsymbol{\epsilon}_n^\top \frac{X_n A_n X_n^\top}{n^2}\Big]\Big) \leq \sigma_\epsilon^2 \mathbb{E}\Big[\mathrm{tr}\Big(\mathbb{E}\Big[\frac{X_n A_n X_n^\top}{n^2}\Big| p\Big]\Big)\Big] = \sigma_\epsilon^2 \mathrm{tr}\Big(\mathbb{E}\Big[\frac{X_n^\top X_n A_n}{n^2}\Big]\Big).$$

Finally, we observe that, thanks to assumption $i)$, we have $\mathbb{E}\big[\epsilon^2\big] = \mathbb{E}\big[\mathbb{E}[\epsilon^2 | p]\big] \leq \sigma_\epsilon^2$. The statement derives from combining the upper bounds on all the terms. ∎

**Remark 9** (Connection to the Mean Estimation Problem). *In Ex. 1, the minimizer of the transfer risk coincides with the mean $\bar{w}$ of the regression vectors. Hence, our problem appears similar to a mean estimation problem (see e.g.[11]). However, in our setting, we do not receive the regression vectors, but we have indirect observations of them by the corresponding datasets. Moreover, in our case, we aim at minimizing the (excess) transfer risk (and not at estimating its minimizer) and, as already observed in Prop. 1, this quantity does not coincide with the quantity $V_h^2 = \frac{1}{2}\mathbb{E}[\|h - \bar{w}\|^2]$. Specifically, also $V_h^2$ is minimized at $h = \bar{w}$, and, for any $h \in \mathbb{R}^d$, in the setting of Ex. 1, we have*

$$\lambda_{\min}(\bar{\Sigma}_n) V_h^2 \leq \mathcal{E}_n(h) - \mathcal{E}_n(h_n) = \mathcal{E}_n(h) - \mathcal{E}_n(\bar{w}) = \frac{1}{2}\big\|\bar{\Sigma}_n^{1/2}(h - \bar{w})\big\|^2 \leq \|\bar{\Sigma}_n\|_\infty V_h^2.$$

**Remark 3** (Advantage of Learning Around the Best Mean over ITL in Ex. 1). *Consider the setting of Ex. 1. If the noise satisfies $\sigma_\epsilon^2 \ll \big(n^{-1}\lambda^{-2} + 1\big)^{-1} \|\bar{\Sigma}_n^{1/2} \bar{w}\|^2$ and the regression vectors are such that $\sigma_w^2 \ll \mathrm{tr}(\bar{\Sigma}_n)^{-1} \|\bar{\Sigma}_n^{1/2} \bar{w}\|^2$, then $\mathcal{E}_n(0) - \mathcal{E}_n(\bar{w}) \gg \mathcal{E}_n(\bar{w})$.*

**Proof.** As already observed in the paper, thanks to Prop. 1, the difference between the transfer risk of the algorithm with $h = 0$ (ITL) and the best algorithm in our class, can be rewritten as $\mathcal{E}_n(0) - \mathcal{E}_n(\bar{w}) = \frac{1}{2}\bar{w}^\top \bar{\Sigma}_n \bar{w}$. Now, in the setting of Ex. 1, we have that

$$\mathcal{E}_n(h) \leq \frac{1}{2}(h - \bar{w})^\top \bar{\Sigma}_n (h - \bar{w}) + \frac{\sigma_w^2 \mathrm{tr}(\bar{\Sigma}_n)}{2} + \frac{\sigma_\epsilon^2}{2} \mathrm{tr}\Big(\mathbb{E}\Big[\frac{X_n^\top X_n A_n}{n^2}\Big]\Big) + \frac{\sigma_\epsilon^2}{2}. \tag{23}$$

Then, the improvement over ITL is significant if $\mathcal{E}_n(0) - \mathcal{E}_n(\bar{w})$ is much greater than the RHS term in Eq. (23) evaluated in $h = \bar{w}$, i.e. if

$$\bar{w}^\top \bar{\Sigma}_n \bar{w} \gg \sigma_w^2 \mathrm{tr}(\bar{\Sigma}_n) + \sigma_\epsilon^2 \mathrm{tr}\Big(\mathbb{E}\Big[\frac{X_n^\top X_n A_n}{n^2}\Big]\Big) + \sigma_\epsilon^2. \tag{24}$$

Now, we observe that, thanks to the assumption $\mathcal{X} \subseteq \mathcal{B}_1$, we have that $\|xx^\top\|_\infty \leq 1$ and $\operatorname{tr}(X_n^\top X_n) = \|X_n\|^2 \leq n$. Hence, exploiting the definition of the matrix $A_n$, the fact that $\|C_{\lambda,n}^{-1}\|_\infty \leq 1/\lambda$ and applying Holder's inequality, we get

$$\operatorname{tr}\Big(\frac{X_n^\top X_n A_n}{n^2}\Big) = \operatorname{tr}\Big(\frac{X_n^\top X_n C_{\lambda,n}^{-1} xx^\top C_{\lambda,n}^{-1}}{n^2}\Big) \leq \frac{\operatorname{tr}(X_n^\top X_n)}{n^2}\|C_{\lambda,n}^{-1} xx^\top C_{\lambda,n}^{-1}\|_\infty \leq n^{-1}\lambda^{-2}.$$

Consequently, instead of analysing Eq. (24), we can simply require that the following inequality

$$\bar{w}^\top \bar{\Sigma}_n \bar{w} \gg \sigma_w^2 \operatorname{tr}(\bar{\Sigma}_n) + \sigma_\epsilon^2 (n^{-1}\lambda^{-2} + 1)$$

holds. In turn, this corresponds to requiring that $\sigma_w^2 \ll \operatorname{tr}(\bar{\Sigma}_n)^{-1}\bar{w}^\top \bar{\Sigma}_n \bar{w} \leq \|\bar{w}\|^2$ and $\sigma_\epsilon^2 \ll (n^{-1}\lambda^{-2} + 1)^{-1}\bar{w}^\top \bar{\Sigma}_n \bar{w} \leq (n^{-1}\lambda^{-2} + 1)^{-1}\|\bar{w}\|^2$, where in the last inequality we have applied Lemma 11 and exploited the fact that, for any $\mu \sim \rho$, thanks to the assumption $\mathcal{X} \subseteq \mathcal{B}_1$, $\|\Sigma_\mu\|_\infty \leq 1$. ∎

As described in the following, the proof of Ex. 2 follows the same lines of the proof of Ex. 1.

**Example 2.** *Let $\mathcal{X} \subseteq \mathcal{B}_1$ and consider the environment $\rho$ formed by $K \in \mathbb{N}\backslash\{0\}$ clusters of tasks parametrized by the triplet $(w, p, \eta)$ as in Asm. 1. Assume that each cluster $k \in [K]$ is associated to a marginal distribution $p_k$ that is sampled with probability $\mathbb{P}(p = p_k) = \nu_k > 0$. For any $k \in [K]$ and $\lambda > 0$, let $\bar{w}_k = \mathbb{E}[w|p = p_k]$, $\bar{\Sigma}_{n,k} = (n\lambda)^2 \mathbb{E}[A_n|p = p_k]$ with $A_n = C_{\lambda,n}^{-1} xx^\top C_{\lambda,n}^{-1}$ and assume that i) $\eta \mid p = p_k$ has variance bounded by $\sigma_{\epsilon,k}^2$ for $\sigma_{\epsilon,k} \geq 0$, ii) $\mathbb{E}[(w - \bar{w}_k)(w - \bar{w}_k)^\top|p = p_k] \preceq \sigma_{w,k}^2 I$ for $\sigma_{w,k} \geq 0$. Then, for any $\lambda > 0$ and $h \in \mathbb{R}^d$, $h_n^* = (\sum_{k=1}^K \nu_k \bar{\Sigma}_{n,k})^{-1}\sum_{k=1}^K \nu_k \bar{\Sigma}_{n,k} \bar{w}_k$ and $\mathcal{E}_n(h) = \sum_{k=1}^K \nu_k \mathcal{E}_{n,k}(h)$, where*

$$\mathcal{E}_{n,k}(h) \leq \frac{1}{2}\big\|\bar{\Sigma}_{n,k}^{1/2}(h - \bar{w}_k)\big\|^2 + \frac{\sigma_{w,k}^2}{2}\operatorname{tr}(\bar{\Sigma}_{n,k}) + \frac{\sigma_{\epsilon,k}^2}{2}\Big(1 + \operatorname{tr}\Big(\mathbb{E}\Big[\frac{X_n^\top X_n A_n}{n^2}\Big|p = p_k\Big]\Big)\Big).$$

**Proof.** We start observing that, since the sampling of the marginal distribution $p$ is drawn from a discrete meta-distribution, exploiting the law of total expectation and using the same notation introduced in the proof of Ex. 1, we can rewrite the transfer risk $\mathcal{E}_n$ of the algorithm $w_h$ on any environment satisfying Asm. 1 as follows

$$\mathcal{E}_n(h) = \frac{1}{2}\mathbb{E}\Big[\big(\langle x, w_h(Z_n)\rangle - y\big)^2\Big] = \frac{1}{2}\sum_{k=1}^K \nu_k \mathbb{E}\Big[\big(\langle x, w_h(Z_n)\rangle - y\big)^2\Big|p = p_k\Big] = \sum_{k=1}^K \nu_k \mathcal{E}_{n,k}(h),$$

where in the last step we have introduced the quantity

$$\mathcal{E}_{n,k}(h) = \frac{1}{2}\mathbb{E}\Big[\big(\langle x, w_h(Z_n)\rangle - y\big)^2\Big|p = p_k\Big].$$

The proof of the closed form of the function $\mathcal{E}_{n,k}(h)$ follows along the same lines of Ex. 1, taking into account the conditioning with respect to $p = p_k$. Since the steps are exactly the same described in the proof Ex. 1, we skip the derivation and we report only the sketch of the reasoning. Specifically, exploiting again the linear model equation $y = \langle x, w\rangle + \epsilon$, Rem. 8 and the fact that the noise $\eta \mid p = p_k$ is zero-mean, we can rewrite

$$\mathcal{E}_{n,k}(h) = \frac{1}{2}\mathbb{E}\Big[(w_h(Z_n) - w)^\top xx^\top (w_h(Z_n) - w)\big|p = p_k\Big] + \frac{\mathbb{E}[\epsilon^2|p = p_k]}{2}.$$

Repeating the same steps in Eqs. (17)-(18)-(19) in the proof of Ex. 1 and denoting by $A_n = C_{\lambda,n}^{-1} xx^\top C_{\lambda,n}^{-1}$, we get

$$\mathcal{E}_{n,k}(h) = \frac{1}{2}\mathbb{E}\Big[\lambda^2(h - w)^\top A_n(h - w) + \frac{\epsilon_n^\top X_n A_n X_n^\top \epsilon_n}{n^2}\Big|p = p_k\Big] + \frac{\mathbb{E}[\epsilon^2|p = p_k]}{2}. \quad (25)$$

Now, letting $w = \bar{w}_k + v_k$, with $v_k \in \mathbb{R}^d$, we can rewrite

$$\mathcal{E}_{n,k}(h) = \frac{1}{2}\mathbb{E}\Big[\lambda^2(h - \bar{w}_k)^\top A_n(h - \bar{w}_k) - 2\lambda^2(h - \bar{w}_k)^\top A_n v_k + \lambda^2 v_k^\top A_n v_k$$
$$+ \frac{\epsilon_n^\top X_n A_n X_n^\top \epsilon_n}{n^2}\Big|p = p_k\Big] + \frac{\mathbb{E}[\epsilon^2|p = p_k]}{2}.$$

We now observe that, thanks to the definition of $\bar{w}_k$, $\mathbb{E}\big[v_k\big|p=p_k\big]=0$, hence, by Rem. 8, we have

$$\mathbb{E}\big[(h-\bar{w}_k)^\top A_n v_k\big|p=p_k\big]=(h-\bar{w}_k)^\top\mathbb{E}\big[A_n v_k\big|p=p_k\big]$$
$$=(h-\bar{w}_k)^\top\mathbb{E}\big[A_n\big|p=p_k\big]\mathbb{E}\big[v_k\big|p=p_k\big]=0.$$

Consequently, exploiting again the relations in Eq. (20) and the definition $\bar{\Sigma}_{n,k}=\mathbb{E}\big[\lambda^2 A_n\big|p=p_k\big]$, we can conclude that

$$\mathcal{E}_{n,k}(h)=\frac{1}{2}(h-\bar{w}_k)^\top\bar{\Sigma}_{n,k}(h-\bar{w}_k)+\frac{1}{2}\mathrm{tr}\big(\mathbb{E}\big[v_k v_k^\top\lambda^2 A_n\big|p=p_k\big]\big)$$
$$+\frac{1}{2}\mathrm{tr}\Big(\mathbb{E}\Big[\boldsymbol{\epsilon}_n\boldsymbol{\epsilon}_n^\top\frac{X_n A_n X_n^\top}{n^2}\Big|p=p_k\Big]\Big)+\frac{\mathbb{E}\big[\epsilon^2\big|p=p_k\big]}{2}.$$

From this last equation, taking the derivative with respect to $h$, we get the closed form of the minimizer $h_n^*$ given in the text. Again, the invertibility of the meta-covariance matrix $\sum_{k=1}^{K}\nu_k\bar{\Sigma}_{n,k}$ is provided by Prop. 1. The upper bound on $\mathcal{E}_{n,k}(h)$ given in the last statement of the example follows by repeating the same steps in the proof of Ex. 1 taking into account the conditioning with respect to $p=p_k$ and exploiting assumptions $i$)-$ii$). ∎

We conclude this section with the proof of Prop. 2

**Proposition 2** (LS Problem Around a Common Mean for $\mathcal{E}_r$). *For any $\lambda>0$, $h\in\mathbb{R}^d$ and $r\in[n-1]$, the transfer risk $\mathcal{E}_r$ in Eq. (7) of the learning algorithm $w_h$ in Eqs. (8)-(9), can be rewritten as*

$$\mathcal{E}_r(h)=\frac{1}{2}\mathbb{E}_{\bar{X}_r,\bar{\mathbf{y}}_r}\Big[\big\|\bar{X}_r h-\bar{\mathbf{y}}_r\big\|^2\Big] \tag{10}$$

*where the meta-data are given by*

$$\bar{X}_r=\frac{\lambda}{\sqrt{n-r}}X_{n-r}C_{\lambda,r}^{-1},\qquad \bar{\mathbf{y}}_r=\frac{1}{\sqrt{n-r}}\Big(\mathbf{y}_{n-r}-\frac{\sqrt{n-r}}{\lambda}\bar{X}_r\frac{X_r^\top\mathbf{y}_r}{n}\Big). \tag{11}$$

*Moreover, under Asm. 1, the meta-covariance matrix $\bar{\Sigma}_r=\mathbb{E}\bar{X}_r^\top\bar{X}_r=\lambda^2\mathbb{E}\big[C_{\lambda,r}^{-1}\Sigma_\mu C_{\lambda,r}^{-1}\big]$ is invertible and $h_r^*=\bar{\Sigma}_r^{-1}\mathbb{E}[\bar{X}_r^\top\bar{\mathbf{y}}_r]$ is the unique minimizer of the LS function in Eq. (10). In such a case, letting $v=w-h_r^*$, we have that $\bar{\mathbf{y}}_r=\bar{X}_r h_r^*+\bar{\boldsymbol{\epsilon}}_r$, with*

$$\bar{\boldsymbol{\epsilon}}_r=\frac{1}{\sqrt{n-r}}\boldsymbol{\epsilon}_{n-r}+\bar{X}_r\Big(v-\frac{X_r^\top\boldsymbol{\epsilon}_r}{\lambda n}\Big). \tag{12}$$

**Proof.** The rewriting of the transfer risk in Eq. (10) is a direct consequence of the closed form of the algorithm in Eq.(9). The invertibility of the meta-covariance matrix is a direct consequence of Lemma 11 for $r\in\{0\}\cup[n-1]$ and the requirement $\mathbb{E}_{\mu\sim\rho}\lambda_{\min}(\Sigma_\mu)>0$ in Asm. 1. This implies that the LS function is strictly convex and, consequently, it admits a unique minimizer $h_r^*$. The closed form of this minimizer directly follows from the optimality conditions (normal equations) of the problem. Now, thanks to Asm. 1, using the linear model equations

$$\mathbf{y}_{n-r}=X_{n-r}w+\boldsymbol{\epsilon}_{n-r}\qquad \mathbf{y}_r=X_r w+\boldsymbol{\epsilon}_r,$$

and letting $w=h_r^*+v$, for some $v\in\mathbb{R}^d$, the following relations hold

$$\bar{\mathbf{y}}_r=\frac{1}{\sqrt{n-r}}\Big(\mathbf{y}_{n-r}-\frac{\sqrt{n-r}}{\lambda}\bar{X}_r\frac{X_r^\top\mathbf{y}_r}{n}\Big)$$
$$=\frac{1}{\sqrt{n-r}}\Big(\mathbf{y}_{n-r}-X_{n-r}C_{\lambda,r}^{-1}\frac{X_r^\top\mathbf{y}_r}{n}\Big)$$
$$=\frac{1}{\sqrt{n-r}}\Big(X_{n-r}w+\boldsymbol{\epsilon}_{n-r}-X_{n-r}C_{\lambda,r}^{-1}\frac{X_r^\top\big(X_r w+\boldsymbol{\epsilon}_r\big)}{n}\Big)$$
$$=\frac{1}{\sqrt{n-r}}\Big(X_{n-r}-X_{n-r}C_{\lambda,r}^{-1}\frac{X_r^\top X_r}{n}\Big)w+\frac{1}{\sqrt{n-r}}\Big(\boldsymbol{\epsilon}_{n-r}-X_{n-r}C_{\lambda,r}^{-1}\frac{X_r^\top\boldsymbol{\epsilon}_r}{n}\Big)$$
$$=\bar{X}_r w+\frac{1}{\sqrt{n-r}}\Big(\boldsymbol{\epsilon}_{n-r}-\frac{\sqrt{n-r}}{\lambda}\bar{X}_r\frac{X_r^\top\boldsymbol{\epsilon}_r}{n}\Big)$$
$$=\bar{X}_r h_r^*+\bar{X}_r v+\frac{1}{\sqrt{n-r}}\Big(\boldsymbol{\epsilon}_{n-r}-\frac{\sqrt{n-r}}{\lambda}\bar{X}_r\frac{X_r^\top\boldsymbol{\epsilon}_r}{n}\Big)$$
$$=\bar{X}_r h_r^*+\frac{1}{\sqrt{n-r}}\boldsymbol{\epsilon}_{n-r}+\bar{X}_r\Big(v-\frac{X_r^\top\boldsymbol{\epsilon}_r}{\lambda n}\Big),$$

where in the fifth equality we have used the fact that

$$\frac{1}{\sqrt{n-r}}\Big(X_{n-r} - X_{n-r}C_{\lambda,r}^{-1}\frac{X_r^\top X_r}{n}\Big) = \frac{1}{\sqrt{n-r}}X_{n-r}C_{\lambda,r}^{-1}\Big(C_{\lambda,r} - \frac{X_r^\top X_r}{n}\Big)$$

$$= \frac{1}{\sqrt{n-r}}X_{n-r}C_{\lambda,r}^{-1}\lambda I = \bar{X}_r.$$

We conclude that the noise on the meta-labels is heteroscedastic, since it is given by

$$\bar{\boldsymbol{\epsilon}}_r = \bar{\mathbf{y}}_r - \bar{X}_r h_r^* = \frac{1}{\sqrt{n-r}}\boldsymbol{\epsilon}_{n-r} + \bar{X}_r\Big(v - \frac{X_r^\top \boldsymbol{\epsilon}_r}{\lambda n}\Big).$$

∎

## C  Least Mean Squares (LMS) Algorithm

In this section, we extend the convergence rate given in [13] for the Least Mean Squares (LMS) algorithm, i.e. SGD applied to a Least Squares problem, when at each iteration we sample a matrix $\bar{X}$ (instead of a single vector as in the standard case) and a vector $\bar{\mathbf{y}}$ (instead of a scalar as in the standard case). The material of this section is essentially based on [13].

### C.1  The problem, the algorithm and the convergence rate

Let $m$ be a positive integer, let $\bar{X} \in \mathbb{R}^{m \times d}$, $\bar{\mathbf{y}} \in \mathbb{R}^m$, and let $\mathcal{D}$ be a distribution for the pair $(\bar{X}, \bar{\mathbf{y}})$. We wish to solve the problem

$$\min_{h \in \mathbb{R}^d} \underbrace{\mathbb{E}_{\bar{Z}=(\bar{X},\bar{\mathbf{y}})\sim\mathcal{D}}\Big[\frac{1}{2}\big\|\bar{X}h - \bar{\mathbf{y}}\big\|^2\Big]}_{f(h)} = \min_{h \in \mathbb{R}^d} f(h). \tag{26}$$

To this end, we assume of having a stream of data $\bar{Z}^{(1)}, \bar{Z}^{(2)}, \ldots$ i.i.d sampled from $\mathcal{D}$ and we apply Alg. 2. In the sequel we will omit the subscript $_{\bar{Z}\sim\mathcal{D}}$ in all the expectations. We now introduce some further notation.

1. The exact covariance matrix (which in the following will be assumed to be invertible):

$$\bar{\Sigma} = \mathbb{E}\big[\bar{X}^{(t)^\top}\bar{X}^{(t)}\big] \in \mathbb{R}^{d \times d}. \tag{27}$$

2. The closed form of the minimizer of the optimization problem in Eq. (26):

$$h^* = \operatorname*{argmin}_{h \in \mathbb{R}^d} f(h) = \bar{\Sigma}^{-1}\mathbb{E}\big[\bar{X}^{(t)^\top}\bar{\mathbf{y}}^{(t)}\big]. \tag{28}$$

3. The residual associated to the seen dataset $\bar{Z}^{(t)}$:

$$\bar{\boldsymbol{\epsilon}}^{(t)} = \bar{\mathbf{y}}^{(t)} - \bar{X}^{(t)}h^* \in \mathbb{R}^m. \tag{29}$$

It follows that

$$\mathbb{E}\big[\bar{X}^{(t)^\top}\bar{\boldsymbol{\epsilon}}^{(t)}\big] = 0. \tag{30}$$

Indeed, exploiting the optimality conditions for $h^*$, i.e. $\bar{\Sigma}h^* = \mathbb{E}\big[\bar{X}^{(t)^\top}\bar{\mathbf{y}}^{(t)}\big]$, we have

$$\mathbb{E}\big[\bar{X}^{(t)^\top}\bar{\boldsymbol{\epsilon}}^{(t)}\big] = \mathbb{E}\big[\bar{X}^{(t)^\top}\big(\bar{\mathbf{y}}^{(t)} - \bar{X}^{(t)}h^*\big)\big] = \mathbb{E}\big[\bar{X}^{(t)^\top}\bar{\mathbf{y}}^{(t)}\big] - \mathbb{E}\big[\bar{X}^{(t)^\top}\bar{X}^{(t)}\big]h^* = 0.$$

In the sequel we will make the following assumptions.

**Assumption 2** (Bounded Inputs). *There exists $R \geq 0$ such that $\big\|\bar{X}^{(t)}\big\|_\infty \leq R$ a.s.*

Asm. 2 implies the following points.

1. Applying Cor. 5 to the matrix $W = \bar{X}^{(t)^\top}\bar{X}^{(t)}$, we get

$$\mathbb{E}\Big[\bar{X}^{(t)^\top}\bar{X}^{(t)}\big(\bar{X}^{(t)^\top}\bar{X}^{(t)}\big)^\top\Big] \preceq \mathbb{E}\Big[\big\|\bar{X}^{(t)^\top}\bar{X}^{(t)}\big\|_\infty\bar{X}^{(t)^\top}\bar{X}^{(t)}\Big] \preceq R^2\bar{\Sigma}. \tag{31}$$

---

**Algorithm 2** SGD (LMS) applied to the function $f$ in Eq. (26)

---

    **Input** $\gamma > 0$ (step size)
    **Initialization** $h^{(0)} \in \mathbb{R}^d$
    **For** $t = 1$ to $T$
        Receive   $\bar{Z}^{(t)} = \left( \bar{X}^{(t)}, \bar{\mathbf{y}}^{(t)} \right)$
        Update   $h^{(t)} = h^{(t-1)} - \gamma \bar{X}^{(t)^\top} \left( \bar{X}^{(t)} h^{(t-1)} - \bar{\mathbf{y}}^{(t)} \right)$
    **Return** $\bar{h}_T = \frac{1}{T+1} \sum_{t=0}^{T} h^{(t)}$

---

2. Thanks to the definition of $\bar{\Sigma}$, we have that

$$\left\| \bar{\Sigma} \right\|_\infty \leq \mathbb{E}\left[ \left\| \bar{X}^{(t)^\top} \bar{X}^{(t)} \right\|_\infty \right] \leq R^2. \tag{32}$$

**Assumption 3** (Noise on the $\bar{\mathbf{y}}$). *There exists $\sigma \geq 0$ such that*

$$\mathbb{E}\left[ \bar{X}^{(t)^\top} \bar{\boldsymbol{\epsilon}}^{(t)} \left( \bar{X}^{(t)^\top} \bar{\boldsymbol{\epsilon}}^{(t)} \right)^\top \right] = \mathbb{E}\left[ \bar{X}^{(t)^\top} \bar{\boldsymbol{\epsilon}}^{(t)} \bar{\boldsymbol{\epsilon}}^{(t)^\top} \bar{X}^{(t)} \right] \preceq \sigma^2 \bar{\Sigma}.$$

**Assumption 4** (Step size $\gamma$). *The step size $\gamma$ is taken such that $\gamma \leq 1/(2R^2)$, where $R^2$ is the constant in Asm. 2.*

**Remark 10.** *Eq. (32) implies that $\gamma \bar{\Sigma} \preceq \gamma \|\bar{\Sigma}\|_\infty I \preceq \gamma R^2 I$. Hence, under Asm. 4, we have that $\gamma \bar{\Sigma} \preceq (1/2)I \prec I$.*

The following result essentially coincides with a simplified version of Thm. 2 in [13].

**Theorem 12.** *Under Asm. 2, Asm. 3 and Asm. 4, let $\bar{h}_T$ be the output of Alg. 2 applied to solve the stochastic LS problem in Eq. (26). Then, $\bar{h}_T$ satisfies the following convergence rate*

$$\mathbb{E}\left[ f(\bar{h}_T) - f(h^*) \right] \leq \frac{2\left\| h^{(0)} - h^* \right\|^2}{\gamma(T+1)} + \frac{2d\sigma^2}{T+1},$$

*where the expectation is over the data $\bar{Z}^{(1)}, \ldots, \bar{Z}^{(T)}$.*

The bound in the above theorem is the sum of two terms: the first one containing the distance between the initial point and the optimal point is the so-called bias term, the second term is the so-called variance term and it represents the variance on the stochastic gradient at the optimal point. We remark also that it is possible to give a faster convergence rate for the bias term, as described in Thm. 2 in [13], however, in order to keep the presentation simple, we avoid this technical step, which is beyond the scope of this work. The following sub-section is devoted to the proof of Thm. 12.

### C.2 Proof of the Convergence Rate

The material of this sub-section is based on App. B in [13]. The starting point for the proof of Thm. 12 is the rewriting of the update rule in Alg. 2 in closed form.

**Lemma 13** (Recursive formula of the update in Alg. 2). *In the setting of Thm. 12, for any positive integers $i, k$ such that $k \leq i + 1$, introduce the operators*

$$M(i,k) = \begin{cases} \left(I - \gamma \bar{X}^{(i)^\top} \bar{X}^{(i)}\right) \cdots \left(I - \gamma \bar{X}^{(k)^\top} \bar{X}^{(k)}\right) & \text{if } i \geq k \\ I & \text{if } k = i + 1. \end{cases} \tag{33}$$

*Then, for any positive integers $i, t$ such that $i \leq t$, we can rewrite*

$$h^{(t)} - h^* = M(t, i+1)(h^{(i)} - h^*) + \gamma \sum_{k=i+1}^{t} M(t, k+1)\bar{X}^{(k)^\top} \bar{\boldsymbol{\epsilon}}^{(k)}. \tag{34}$$

*In particular, for $i = 0$, we get*

$$h^{(t)} - h^* = M(t,1)(h^{(0)} - h^*) + \gamma \sum_{k=1}^{t} M(t, k+1)\bar{X}^{(k)^\top} \bar{\boldsymbol{\epsilon}}^{(k)}. \tag{35}$$

**Proof.** Looking at the update step in Alg. 2, using the fact $\bar{\mathbf{y}}^{(t)} = \bar{\boldsymbol{\epsilon}}^{(t)} + \bar{X}^{(t)} h^*$, we can rewrite

$$h^{(t)} - h^* = \big(I - \gamma \bar{X}^{(t)^\top} \bar{X}^{(t)}\big)(h^{(t-1)} - h^*) + \gamma \bar{X}^{(t)^\top} \bar{\boldsymbol{\epsilon}}^{(t)}. \tag{36}$$

Iterating Eq. (36) over $t$ and using the definition in Eq. (33), we get the statement. ∎

**Remark 11.** *We observe that the operator $M(i,k)$ defined in Eq. (33) depends only on $\bar{X}^{(k)}, \ldots, \bar{X}^{(i)}$, moreover, since the points are i.i.d., we have that*

$$\mathbb{E}\big[M(i,k)\big] = \big(I - \gamma \bar{\Sigma}\big)^{i-k+1}. \tag{37}$$

The proof of the convergence rate in Thm. 12 relies on the bias-variance decomposition described in the following proposition.

**Proposition 14** (Bias-Variance Decomposition)**.** *Under the assumptions of Thm. 12, we have that*

$$\mathbb{E}\big[f(\bar{h}_T) - f(h^*)\big] \leq \boldsymbol{V} + \boldsymbol{B},$$

*where the expectation is over the data $\bar{Z}^{(1)}, \ldots, \bar{Z}^{(T)}$ and we have introduced the bias and the variance terms, respectively given by*

$$\boldsymbol{B} = \frac{2}{\gamma(T+1)^2} \sum_{t=0}^{T} \mathbb{E}\Big[\big\|M(t,1)(h^{(0)} - h^*)\big\|^2\Big]$$

$$\boldsymbol{V} = \frac{2\gamma}{(T+1)^2} \sum_{t=0}^{T} \mathbb{E}\Big[\Big\|\sum_{k=1}^{t} M(t,k+1)\bar{X}^{(k)^\top} \bar{\boldsymbol{\epsilon}}^{(k)}\Big\|^2\Big].$$

**Proof.** As well known in Least Squares theory, the starting point of the proof is to exploit the equality

$$f(\bar{h}_T) - f(h^*) = \frac{1}{2}\big\|\bar{\Sigma}^{1/2}(\bar{h}_T - h^*)\big\|^2.$$

We now observe that

$$
\begin{aligned}
(T+1)^2 \big\|\bar{\Sigma}^{1/2}(\bar{h}_T - h^*)\big\|^2 &= (T+1)^2 \Big\langle \bar{h}_T - h^*, \bar{\Sigma}(\bar{h}_T - h^*)\Big\rangle \\
&= \sum_{t=0}^{T} \sum_{j=0}^{T} \Big\langle h^{(t)} - h^*, \bar{\Sigma}(h^{(j)} - h^*)\Big\rangle \\
&= \sum_{t=0}^{T} \Big\langle h^{(t)} - h^*, \bar{\Sigma}(h^{(t)} - h^*)\Big\rangle + 2\sum_{t=0}^{T-1} \sum_{j=t+1}^{T} \Big\langle h^{(t)} - h^*, \bar{\Sigma}(h^{(j)} - h^*)\Big\rangle \\
&= \sum_{t=0}^{T} \big\|\bar{\Sigma}^{1/2}(h^{(t)} - h^*)\big\|^2 + 2\sum_{t=0}^{T-1} \sum_{j=t+1}^{T} \Big\langle h^{(t)} - h^*, \bar{\Sigma}(h^{(j)} - h^*)\Big\rangle,
\end{aligned}
$$

where in the third equality we have applied Lemma 10 to $i_{\min} = 0$, $i_{\max} = T$ and $a_{t,j} = \big\langle h^{(t)} - h^*, \bar{\Sigma}(h^{(j)} - h^*)\big\rangle$. Hence, taking the expectation over the data $\bar{Z}^{(1)}, \ldots, \bar{Z}^{(T)}$, we get

$$
\begin{aligned}
(T+1)^2 \mathbb{E}\Big[\big\|\bar{\Sigma}^{1/2}(\bar{h}_T - h^*)\big\|^2\Big] &= \sum_{t=0}^{T} \mathbb{E}\Big[\big\|\bar{\Sigma}^{1/2}(h^{(t)} - h^*)\big\|^2\Big] \\
&\quad + 2\underbrace{\sum_{t=0}^{T-1} \sum_{j=t+1}^{T} \mathbb{E}\Big\langle h^{(t)} - h^*, \bar{\Sigma}(h^{(j)} - h^*)\Big\rangle}_{C}.
\end{aligned} \tag{38}
$$

Using the recursive formula in Eq. (34), more precisely

$$h^{(j)} - h^* = M(j,t+1)(h^{(t)} - h^*) + \gamma \sum_{k=t+1}^{j} M(j,k+1)\bar{X}^{(k)^\top} \bar{\boldsymbol{\epsilon}}^{(k)},$$

we can write the term C as follows.

$$
\begin{aligned}
C &= \sum_{t=0}^{T-1} \sum_{j=t+1}^{T} \mathbb{E}\Big\langle h^{(t)} - h^*, \bar{\Sigma}(h^{(j)} - h^*)\Big\rangle \\
&= \sum_{t=0}^{T-1} \sum_{j=t+1}^{T} \mathbb{E}\Big\langle h^{(t)} - h^*, \bar{\Sigma}\Big(M(j,t+1)(h^{(t)} - h^*) + \gamma \sum_{k=t+1}^{j} M(j,k+1)\bar{X}^{(k)^\top} \bar{\epsilon}^{(k)}\Big)\Big\rangle \\
&= \sum_{t=0}^{T-1} \sum_{j=t+1}^{T} \mathbb{E}\Big\langle h^{(t)} - h^*, \bar{\Sigma}M(j,t+1)(h^{(t)} - h^*)\Big\rangle \\
&\qquad + \gamma \sum_{t=0}^{T-1} \sum_{j=t+1}^{T} \sum_{k=t+1}^{j} \mathbb{E}\Big\langle h^{(t)} - h^*, \bar{\Sigma}M(j,k+1)\bar{X}^{(k)^\top} \bar{\epsilon}^{(k)}\Big\rangle.
\end{aligned}
$$

We now observe that, thanks to the constraints we have on the indexes $t, j, k$ and thanks to the independence of the sampled points, the variables $h^{(t)}$, $M(j, k+1)$ and $\bar{X}^{(k)^\top} \bar{\epsilon}^{(k)}$ are independent. Consequently, since $\mathbb{E}\big[\bar{X}^{(k)^\top} \bar{\epsilon}^{(k)}\big] = 0$, the second term vanishes. Hence, exploiting again the independence of $h^{(t)}$ and $M(j, t+1)$, we have that

$$
\begin{aligned}
C &= \sum_{t=0}^{T-1} \sum_{j=t+1}^{T} \mathbb{E}\Big\langle h^{(t)} - h^*, \bar{\Sigma}M(j,t+1)(h^{(t)} - h^*)\Big\rangle \\
&= \sum_{t=0}^{T-1} \sum_{j=t+1}^{T} \mathbb{E}\Big\langle h^{(t)} - h^*, \bar{\Sigma}\mathbb{E}\big[M(j,t+1)\big](h^{(t)} - h^*)\Big\rangle \\
&= \sum_{t=0}^{T-1} \sum_{j=t+1}^{T} \mathbb{E}\Big\langle h^{(t)} - h^*, \bar{\Sigma}\big(I - \gamma\bar{\Sigma}\big)^{j-t}(h^{(t)} - h^*)\Big\rangle \\
&= \sum_{t=0}^{T-1} \mathbb{E}\Big\langle h^{(t)} - h^*, \bar{\Sigma} \sum_{j=t+1}^{T} \big(I - \gamma\bar{\Sigma}\big)^{j-t}(h^{(t)} - h^*)\Big\rangle \\
&= \sum_{t=0}^{T-1} \mathbb{E}\Big\langle h^{(t)} - h^*, \bar{\Sigma} \sum_{k=1}^{T-t} \big(I - \gamma\bar{\Sigma}\big)^{k}(h^{(t)} - h^*)\Big\rangle \\
&\leq \sum_{t=0}^{T-1} \mathbb{E}\Big\langle h^{(t)} - h^*, \bar{\Sigma} \sum_{k=1}^{\infty} \big(I - \gamma\bar{\Sigma}\big)^{k}(h^{(t)} - h^*)\Big\rangle,
\end{aligned} \tag{39}
$$

where in the third equality we have used Eq. (37), in the last equality we have made the change of indexes $k = j - t$ and in the last inequality, we have exploited the fact that we are adding positive quantities. As a matter of fact, for any $k$, the matrices $\bar{\Sigma}\big(I - \gamma\bar{\Sigma}\big)^{k} \in \mathbb{S}_+^d$ (we apply Lemma 6 to $A = \bar{\Sigma} \in \mathbb{S}_{++}^d$ and $B = I - \gamma\bar{\Sigma} \in \mathbb{S}_{++}^d$, thanks to Rem. 10). Now, thanks to Rem. 10 and the invertibility of $\bar{\Sigma}$, we we can apply Lemma 9 to the matrix $A = \gamma\bar{\Sigma}$, hence, we can write

$$
\sum_{k=1}^{\infty} \big(I - \gamma\bar{\Sigma}\big)^{k} = \gamma^{-1}\bar{\Sigma}^{-1}\big(I - \gamma\bar{\Sigma}\big).
$$

Consequently, multiplying by $\bar{\Sigma}$, we obtain

$$
\bar{\Sigma} \sum_{k=1}^{\infty} \big(I - \gamma\bar{\Sigma}\big)^{k} = \gamma^{-1}\big(I - \gamma\bar{\Sigma}\big).
$$

Coming back to Eq. (39), since, as already observed, $I - \gamma \bar{\Sigma} \succ 0$, we get

$$
\begin{aligned}
C &\leq \gamma^{-1} \sum_{t=0}^{T-1} \mathbb{E}\Big\langle h^{(t)} - h^*, \big(I - \gamma \bar{\Sigma}\big)(h^{(t)} - h^*) \Big\rangle \\
&\leq \gamma^{-1} \sum_{t=0}^{T} \mathbb{E}\Big\langle h^{(t)} - h^*, \big(I - \gamma \bar{\Sigma}\big)(h^{(t)} - h^*) \Big\rangle \\
&= \gamma^{-1} \sum_{t=0}^{T} \mathbb{E}\Big[ \big\| h^{(t)} - h^* \big\|^2 \Big] - \sum_{t=0}^{T} \mathbb{E}\Big[ \big\| \bar{\Sigma}^{1/2}(h^{(t)} - h^*) \big\|^2 \Big].
\end{aligned}
$$

Continuing with Eq. (38), we get

$$
\begin{aligned}
(T+1)^2 &\mathbb{E}\Big[ \big\| \bar{\Sigma}^{1/2}(\bar{h}_T - h^*) \big\|^2 \Big] \\
&= \sum_{t=0}^{T} \mathbb{E}\Big[ \big\| \bar{\Sigma}^{1/2}(h^{(t)} - h^*) \big\|^2 \Big] + 2 \underbrace{\sum_{t=0}^{T-1} \sum_{j=t+1}^{T} \mathbb{E}\Big\langle h^{(t)} - h^*, \bar{\Sigma}(h_j - h^*) \Big\rangle}_{C} \\
&\leq \sum_{t=0}^{T} \mathbb{E}\Big[ \big\| \bar{\Sigma}^{1/2}(h^{(t)} - h^*) \big\|^2 \Big] + 2\gamma^{-1} \sum_{t=0}^{T} \mathbb{E}\Big[ \big\| h^{(t)} - h^* \big\|^2 \Big] \\
&\quad - 2 \sum_{t=0}^{T} \mathbb{E}\Big[ \big\| \bar{\Sigma}^{1/2}(h^{(t)} - h^*) \big\|^2 \Big] \\
&= 2\gamma^{-1} \sum_{t=0}^{T} \mathbb{E}\Big[ \big\| h^{(t)} - h^* \big\|^2 \Big] - \sum_{t=0}^{T} \mathbb{E}\Big[ \big\| \bar{\Sigma}^{1/2}(h^{(t)} - h^*) \big\|^2 \Big] \\
&\leq 2\gamma^{-1} \sum_{t=0}^{T} \mathbb{E}\Big[ \big\| h^{(t)} - h^* \big\|^2 \Big].
\end{aligned}
$$

To sum up we have obtained that

$$
\frac{1}{2} \mathbb{E}\Big[ \big\| \bar{\Sigma}^{1/2}(\bar{h}_T - h^*) \big\|^2 \Big] \leq \frac{1}{\gamma(T+1)^2} \sum_{t=0}^{T} \mathbb{E}\Big[ \big\| h^{(t)} - h^* \big\|^2 \Big]. \tag{40}
$$

Now, thanks to Eq. (35), we have that

$$
h^{(t)} - h^* = \underbrace{M(t,1)(h^{(0)} - h^*)}_{A_t} + \underbrace{\gamma \sum_{k=1}^{t} M(t, k+1) \bar{X}^{(k)\top} \bar{\epsilon}^{(k)}}_{B_t},
$$

hence, using Minkowski's inequality, namely for any two random vectors $v_1, v_2$:

$$
\mathbb{E}\big[ \| v_1 + v_2 \|^2 \big] \leq \left( \sqrt{\mathbb{E}\big[ \| v_1 \|^2 \big]} + \sqrt{\mathbb{E}\big[ \| v_2 \|^2 \big]} \right)^2 \leq 2\Big( \mathbb{E}\big[ \| v_1 \|^2 \big] + \mathbb{E}\big[ \| v_2 \|^2 \big] \Big),
$$

we get

$$
\mathbb{E}\Big[ \big\| h^{(t)} - h^* \big\|^2 \Big] = \mathbb{E}\Big[ \| A_t + B_t \|^2 \Big] \leq 2\Big( \mathbb{E}\Big[ \| A_t \|^2 \Big] + \mathbb{E}\Big[ \| B_t \|^2 \Big] \Big).
$$

Returning to Eq. (40), we get

$$
\begin{aligned}
\frac{1}{2}\mathbb{E}\Big[\big\|\bar{\Sigma}^{1/2}(\bar{h}_T - h^*)\big\|^2\Big] &\leq \frac{2}{\gamma(T+1)^2}\sum_{t=0}^{T}\Big(\mathbb{E}\big[\|A_t\|^2\big] + \mathbb{E}\big[\|B_t\|^2\big]\Big) \\
&= \underbrace{\frac{2}{\gamma(T+1)^2}\sum_{t=0}^{T}\mathbb{E}\Big[\big\|M(t,1)(h^{(0)} - h^*)\big\|^2\Big]}_{\mathbf{B}} \\
&\quad + \underbrace{\frac{2\gamma}{(T+1)^2}\sum_{t=0}^{T}\mathbb{E}\Big[\big\|\sum_{k=1}^{t}M(t,k+1)\bar{X}^{(k)\top}\bar{\boldsymbol{\epsilon}}^{(k)}\big\|^2\Big]}_{\mathbf{V}}.
\end{aligned}
\tag{41}
$$

■

In the sequel we give a bound on the separate terms **B** and **V**. We start now from the term **B**.

**Lemma 15** (Bound on the Bias Term **B**). *Under the assumptions of Thm. 12, we have that*

$$
\mathbf{B} = \frac{2}{\gamma(T+1)^2}\sum_{t=0}^{T}\mathbb{E}\Big[\big\|M(t,1)(h^{(0)} - h^*)\big\|^2\Big] \leq \frac{2\|h^{(0)} - h^*\|^2}{\gamma(T+1)}.
\tag{42}
$$

**Proof.** We start observing that, given $A, B$ matrices and $v, w$ vectors, we have that

$$
\langle Av, Bw\rangle = \operatorname{tr}\big((Av)^\top Bw\big) = \operatorname{tr}\big(v^\top A^\top Bw\big) = \operatorname{tr}\big(A^\top Bwv^\top\big).
\tag{43}
$$

Hence, introducing the notation $E^{(0)} = (h^{(0)} - h^*)(h^{(0)} - h^*)^\top$, we can rewrite

$$
\begin{aligned}
\sum_{t=0}^{T}\mathbb{E}\Big[\big\|M(t,1)(h^{(0)} - h^*)\big\|^2\Big] &= \sum_{t=0}^{T}\mathbb{E}\Big\langle \underbrace{M(t,1)}_{A}\underbrace{(h^{(0)} - h^*)}_{v}, \underbrace{M(t,1)}_{B}\underbrace{(h^{(0)} - h^*)}_{w}\Big\rangle \\
&= \mathbb{E}\Big[\sum_{t=0}^{T}\operatorname{tr}\Big(M(t,1)^\top M(t,1)E^{(0)}\Big)\Big] \\
&= \operatorname{tr}\Big(\sum_{t=0}^{T}\mathbb{E}\Big[M(t,1)^\top M(t,1)\Big]E^{(0)}\Big) \\
&= \operatorname{tr}\Big(E^{(0)\,1/2}\sum_{t=0}^{T}\mathbb{E}\Big[M(t,1)^\top M(t,1)\Big]E^{(0)\,1/2}\Big).
\end{aligned}
\tag{44}
$$

Now we observe that

$$
\mathbb{E}\Big[M(t,1)^\top M(t,1)\Big] \prec I,
\tag{45}
$$

as a matter of fact, exploiting again the independence of the points, according to the definition in Eq. (33), we have that

$$
\mathbb{E}\Big[M(t,1)^\top M(t,1)\Big] = \mathbb{E}\Big[M(t-1,1)^\top \mathbb{E}\Big[\big(I - \gamma\bar{X}^{(t)\top}\bar{X}^{(t)}\big)^\top\big(I - \gamma\bar{X}^{(t)\top}\bar{X}^{(t)}\big)\Big]M(t-1,1)\Big]
$$

and, using Eq. (31) and Asm. 4, according to which $2 - \gamma R^2 \geq 3/2 > 0$, we get

$$
\begin{aligned}
\mathbb{E}\Big[\big(I - \gamma\bar{X}^{(t)\top}\bar{X}^{(t)}\big)^\top\big(I - \gamma\bar{X}^{(t)\top}\bar{X}^{(t)}\big)\Big] &= \mathbb{E}\Big[I - 2\gamma\bar{X}^{(t)\top}\bar{X}^{(t)} + \gamma^2\bar{X}^{(t)\top}\bar{X}^{(t)}\bar{X}^{(t)\top}\bar{X}^{(t)}\Big] \\
&\preceq I - 2\gamma\bar{\Sigma} + \gamma^2 R^2\bar{\Sigma} \\
&= I - \gamma\big(2 - \gamma R^2\big)\bar{\Sigma} \prec I.
\end{aligned}
$$

Iterating the previous observation and exploiting Lemma 4 with

$$
\begin{aligned}
U &= I - \mathbb{E}\Big[\big(I - \gamma\bar{X}^{(t)\top}\bar{X}^{(t)}\big)^\top\big(I - \gamma\bar{X}^{(t)\top}\bar{X}^{(t)}\big)\Big] \\
V &= M(t-1,1),
\end{aligned}
$$

we get Eq. (45). Hence, coming back to Eq. (44), applying Lemma 4 with

$$U = I - \mathbb{E}\Big[M(t,1)^{\top}M(t,1)\Big]$$
$$V = E^{(0)\,1/2},$$

we get

$$\sum_{t=0}^{T} \mathbb{E}\Big[\big\|M(t,1)(h^{(0)} - h^*)\big\|^2\Big] \leq (T+1)\mathrm{tr}\big(E^{(0)}\big) = (T+1)\big\|h^{(0)} - h^*\big\|^2.$$

Therefore, for the term **B**, we get the following upper bound:

$$\mathbf{B} = \frac{2}{\gamma(T+1)^2}\sum_{t=0}^{T} \mathbb{E}\Big[\big\|M(t,1)(h^{(0)} - h^*)\big\|^2\Big] \leq \frac{2\big\|h^{(0)} - h^*\big\|^2}{\gamma(T+1)}.$$

∎

In order to give a bound on the variance term **V**, we will exploit the following lemma.

**Lemma 16** (Recursive formula for $M(t, k+1)$)**.** *Under the assumptions of Thm. 12, we have that*

$$\mathbb{E}\Big[M(t,k+1)\bar{\Sigma}M(t,k+1)^{\top}\Big] \preceq$$
$$\frac{1}{\gamma\big(2-\gamma R^2\big)}\Big(\mathbb{E}\Big[M(t,k+1)M(t,k+1)^{\top}\Big] - \mathbb{E}\Big[M(t,k)M(t,k)^{\top}\Big]\Big).$$

**Proof.** Exploiting the independence of the points, we have that:

$$\mathbb{E}\Big[M(t,k)M(t,k)^{\top}\Big]$$
$$= \mathbb{E}\Big[M(t,k+1)\mathbb{E}\Big[\big(I - \gamma\bar{X}^{(k)\top}\bar{X}^{(k)}\big)\big(I - \gamma\bar{X}^{(k)\top}\bar{X}^{(k)}\big)^{\top}\Big]M(t,k+1)^{\top}\Big]$$
$$= \mathbb{E}\Big[M(t,k+1)\Big(I - 2\gamma\bar{\Sigma} + \gamma^2\mathbb{E}\big[\bar{X}^{(k)\top}\bar{X}^{(k)}\bar{X}^{(k)\top}\bar{X}^{(k)}\big]\Big)M(t,k+1)^{\top}\Big] \tag{46}$$
$$= \mathbb{E}\Big[M(t,k+1)M(t,k+1)^{\top}\Big]$$
$$\qquad + \gamma\mathbb{E}\Big[M(t,k+1)\Big(-2\bar{\Sigma} + \gamma\mathbb{E}\big[\bar{X}^{(k)\top}\bar{X}^{(k)}\bar{X}^{(k)\top}\bar{X}^{(k)}\big]\Big)M(t,k+1)^{\top}\Big].$$

But, since, because of Eq. (31), we have that

$$-2\bar{\Sigma} + \gamma\mathbb{E}\big[\bar{X}^{(k)\top}\bar{X}^{(k)}\bar{X}^{(k)\top}\bar{X}^{(k)}\big] \preceq \big(\gamma R^2 - 2\big)\bar{\Sigma} = -\big(2 - \gamma R^2\big)\bar{\Sigma},$$

then, substituting in Eq. (46) and exploiting Lemma 4 with

$$U = -\big(2 - \gamma R^2\big)\bar{\Sigma} - \Big(-2\bar{\Sigma} + \gamma\mathbb{E}\big[\bar{X}^{(k)\top}\bar{X}^{(k)}\bar{X}^{(k)\top}\bar{X}^{(k)}\big]\Big)$$
$$V = M(t,k+1)^{\top},$$

we get

$$\mathbb{E}\Big[M(t,k)M(t,k)^{\top}\Big] \preceq \mathbb{E}\Big[M(t,k+1)M(t,k+1)^{\top}\Big]$$
$$\qquad - \gamma\big(2 - \gamma R^2\big)\mathbb{E}\Big[M(t,k+1)\bar{\Sigma}M(t,k+1)^{\top}\Big].$$

Consequently, since, as already observed, thanks to Asm. 4, we have that $2 - \gamma R^2 \geq 3/2 > 0$, we get

$$\mathbb{E}\Big[M(t,k+1)\bar{\Sigma}M(t,k+1)^{\top}\Big] \preceq$$
$$\frac{1}{\gamma\big(2-\gamma R^2\big)}\Big(\mathbb{E}\Big[M(t,k+1)M(t,k+1)^{\top}\Big] - \mathbb{E}\Big[M(t,k)M(t,k)^{\top}\Big]\Big).$$

∎

We now are ready to give a bound on the term **V**.

**Lemma 17** (Bound on the Variance Term **V**). *Under the assumptions of Thm. 12, we have that*

$$\boldsymbol{V} = \frac{2\gamma}{(T+1)^2} \sum_{t=0}^{T} \mathbb{E}\Big[\Big\| \sum_{k=1}^{t} M(t, k+1) \bar{X}^{(k)^\top} \bar{\boldsymbol{\epsilon}}^{(k)} \Big\|^2\Big] \le \frac{2d\sigma^2}{T+1}. \tag{47}$$

**Proof.** We start observing that applying Lemma 10 to $i_{\min} = 1$, $i_{\max} = t$ and

$$a_{k,j} = \big(\bar{X}^{(j)^\top} \bar{\boldsymbol{\epsilon}}^{(j)}\big)^\top M(t, j+1)^\top M(t, k+1) \bar{X}^{(k)^\top} \bar{\boldsymbol{\epsilon}}^{(k)},$$

we can rewrite

$$\mathbb{E}\Big[\Big\| \sum_{k=1}^{t} M(t, k+1) \bar{X}^{(k)^\top} \bar{\boldsymbol{\epsilon}}^{(k)} \Big\|^2\Big] = \mathbb{E}\Big[ \sum_{k=1}^{t} \sum_{j=1}^{t} \big(\bar{X}^{(j)^\top} \bar{\boldsymbol{\epsilon}}^{(j)}\big)^\top M(t, j+1)^\top M(t, k+1) \bar{X}^{(k)^\top} \bar{\boldsymbol{\epsilon}}^{(k)} \Big]$$

$$= \mathbb{E}\Big[ \sum_{k=1}^{t} \big(\bar{X}^{(k)^\top} \bar{\boldsymbol{\epsilon}}^{(k)}\big)^\top M(t, k+1)^\top M(t, k+1) \bar{X}^{(k)^\top} \bar{\boldsymbol{\epsilon}}^{(k)} \Big]$$

$$+ 2\mathbb{E}\Big[ \sum_{k=1}^{t-1} \sum_{j=k+1}^{t} \big(\bar{X}^{(j)^\top} \bar{\boldsymbol{\epsilon}}^{(j)}\big)^\top M(t, j+1)^\top M(t, k+1) \bar{X}^{(k)^\top} \bar{\boldsymbol{\epsilon}}^{(k)} \Big].$$

But, thanks to the independence of the points and the constraints on the indexes, since $\big(\bar{X}^{(j)^\top} \bar{\boldsymbol{\epsilon}}^{(j)}\big)^\top M(t, j+1)^\top M(t, k+1)$ does not depend on $\bar{X}^{(k)}$ and since $\mathbb{E}\big[\bar{X}^{(k)^\top} \bar{\boldsymbol{\epsilon}}^{(k)}\big] = 0$, we have that

$$\mathbb{E}\Big[ \sum_{k=1}^{t-1} \sum_{j=k+1}^{t} \big(\bar{X}^{(j)^\top} \bar{\boldsymbol{\epsilon}}^{(j)}\big)^\top M(t, j+1)^\top M(t, k+1) \bar{X}^{(k)^\top} \bar{\boldsymbol{\epsilon}}^{(k)} \Big] = 0.$$

Consequently, we can write the following steps.

$$\mathbb{E}\Big[\Big\| \sum_{k=1}^{t} M(t, k+1) \bar{X}^{(k)^\top} \bar{\boldsymbol{\epsilon}}^{(k)} \Big\|^2\Big] = \mathbb{E}\Big[ \sum_{k=1}^{t} \big(\bar{X}^{(k)^\top} \bar{\boldsymbol{\epsilon}}^{(k)}\big)^\top M(t, k+1)^\top M(t, k+1) \bar{X}^{(k)^\top} \bar{\boldsymbol{\epsilon}}^{(k)} \Big]$$

$$= \mathbb{E}\Big[ \sum_{k=1}^{t} \text{tr}\Big( \big(\bar{X}^{(k)^\top} \bar{\boldsymbol{\epsilon}}^{(k)}\big)^\top M(t, k+1)^\top M(t, k+1) \bar{X}^{(k)^\top} \bar{\boldsymbol{\epsilon}}^{(k)} \Big) \Big]$$

$$= \mathbb{E}\Big[ \sum_{k=1}^{t} \text{tr}\Big( \bar{X}^{(k)^\top} \bar{\boldsymbol{\epsilon}}^{(k)} \big(\bar{X}^{(k)^\top} \bar{\boldsymbol{\epsilon}}^{(k)}\big)^\top M(t, k+1)^\top M(t, k+1) \Big) \Big]$$

$$= \text{tr}\Big( \sum_{k=1}^{t} \mathbb{E}\Big[ \bar{X}^{(k)^\top} \bar{\boldsymbol{\epsilon}}^{(k)} \big(\bar{X}^{(k)^\top} \bar{\boldsymbol{\epsilon}}^{(k)}\big)^\top \Big] \mathbb{E}\Big[ M(t, k+1)^\top M(t, k+1) \Big] \Big)$$

$$\le \sigma^2 \text{tr}\Big( \sum_{k=1}^{t} \bar{\Sigma} \mathbb{E}\Big[ M(t, k+1)^\top M(t, k+1) \Big] \Big)$$

$$= \sigma^2 \text{tr}\Big( \sum_{k=1}^{t} \mathbb{E}\Big[ \bar{\Sigma} M(t, k+1)^\top M(t, k+1) \Big] \Big)$$

$$= \sigma^2 \text{tr}\Big( \sum_{k=1}^{t} \mathbb{E}\Big[ M(t, k+1) \bar{\Sigma} M(t, k+1)^\top \Big] \Big), \tag{48}$$

where, in the second equality we have used Eq. (43) and in the fourth equality we have exploited the independence of the points and the definition in Eq. (33). Finally, in the above inequality we have applied Asm. 3 and Lemma 4 with

$$U = \sigma^2 \bar{\Sigma} - \mathbb{E}\Big[ \bar{X}^{(k)^\top} \bar{\boldsymbol{\epsilon}}^{(k)} \big(\bar{X}^{(k)^\top} \bar{\boldsymbol{\epsilon}}^{(k)}\big)^\top \Big]$$

$$V = \mathbb{E}\Big[ M(t, k+1)^\top M(t, k+1) \Big]^{1/2}.$$

Now we observe that, exploiting the recursive formula in Lemma 16 and the fact that we obtain a telescopic sum, we have that

$$\sum_{k=1}^{t} \mathbb{E}\Big[M(t,k+1)\bar{\Sigma}M(t,k+1)^{\top}\Big]$$

$$\preceq \frac{1}{\gamma(2-\gamma R^2)}\sum_{k=1}^{t}\Big(\mathbb{E}\Big[M(t,k+1)M(t,k+1)^{\top}\Big] - \mathbb{E}\Big[M(t,k)M(t,k)^{\top}\Big]\Big)$$

$$= \frac{1}{\gamma(2-\gamma R^2)}\Big(\mathbb{E}\Big[M(t,t+1)M(t,t+1)^{\top}\Big] - \mathbb{E}\Big[M(t,1)M(t,1)^{\top}\Big]\Big)$$

$$\preceq \frac{1}{\gamma(2-\gamma R^2)}I,$$

where in the last step we have used the definition $M(t,t+1) = I$ and the fact $\mathbb{E}\Big[M(t,1)M(t,1)^{\top}\Big] \succeq 0$. Hence, coming back to Eq. (48), we get

$$\mathbb{E}\Big[\Big\|\sum_{k=1}^{t}M(t,k+1)\bar{X}_k^{\top}\bar{\epsilon}_k\Big\|^2\Big] \leq \frac{\sigma^2\mathrm{tr}(I)}{\gamma(2-\gamma R^2)} = \frac{d\sigma^2}{\gamma(2-\gamma R^2)}.$$

Substituting in the definition of **V**, we get the following upper bound.

$$\mathbf{V} \leq \frac{2d\sigma^2}{(2-\gamma R^2)(T+1)}.$$

Since $1/(2-\gamma R^2) < 1$ (as already observed, thanks to Asm. 4, $2-\gamma R^2 \geq 3/2 > 0$), we finally obtain that

$$\mathbf{V} \leq \frac{2d\sigma^2}{T+1}.$$

∎

Finally, we have all the ingredients necessary to prove Thm. 12.

**Proof of Thm. 12.** The statement of the theorem directly follows from combining Prop. 14 with Lemma 15 and Lemma 17. ∎

## D  Adaptation of LMS to the Problem of Minimizing $\mathcal{E}_r$

In this section we adapt the theory described in the previous section to the stochastic problem of minimizing the transfer risk $\mathcal{E}_r$ ($r \in \{0\} \cup [n-1]$) described in the paper. More precisely, when the environment satisfies the assumptions described in Ex. 1, we are able to explicitly estimate the constant $\sigma$ introduced in Asm. 3 and to derive a consequent version of the rate in Thm. 12 for Alg. 1, coinciding with SGD applied to $\mathcal{E}_r$. Such convergence rate will be used in the next App. E, to prove the statement in Prop. 3. Starting from now, we use again the notation introduced in the paper.

For any $r \in \{0\} \cup [n-1]$, the problem of minimizing the function $\mathcal{E}_r$ introduced in Prop. 2 in the paper can be written as in Eq. (26) making the following identifications:

$$m \mapsto n-r$$
$$\bar{X} \mapsto \bar{X}_r = \frac{\lambda}{\sqrt{n-r}}X_{n-r}C_{\lambda,r}^{-1} \in \mathbb{R}^{(n-r)\times d}$$
$$\bar{\mathbf{y}} \mapsto \bar{\mathbf{y}}_r = \frac{1}{\sqrt{n-r}}\Big(\mathbf{y}_{n-r} - \frac{\sqrt{n-r}}{\lambda}\bar{X}_r\frac{X_r^{\top}\mathbf{y}_r}{n}\Big) \in \mathbb{R}^{n-r} \qquad (49)$$
$$\bar{\epsilon} \mapsto \bar{\epsilon}_r = \frac{1}{\sqrt{n-r}}\epsilon_{n-r} + \bar{X}_r\Big(v - \frac{X_r^{\top}\epsilon_r}{\lambda n}\Big) \in \mathbb{R}^{n-r}$$
$$\bar{\Sigma} \mapsto \bar{\Sigma}_r \in \mathbb{R}^{d\times d},$$

where we recall that $v = w - h_r^*$ and the remaining quantities are introduced in Prop. 2. Moreover, the sampling of the points from the distribution $\mathcal{D}$ coincides, in our case, with the sampling of the meta-datasets $(\bar{X}_r, \bar{\mathbf{y}}_r)$ from the distribution induced by the independent sampling of the original datasets $(X_n, \mathbf{y}_n)$ from the environment. Hence, in our setting, the meta-points are independently sampled from this induced distribution. In order to give a convergence rate for Alg. 1, we now specialize all the assumptions introduced in App. C to the setting described in the paper. We start from Asm. 2.

**Lemma 18** (Asm. 2 for $\mathcal{E}_r$). *Consider the setting described in the paper and outlined in Eq. (49) and assume $\mathcal{X} \subseteq \mathcal{B}_1$. Then, for any $r \in \{0\} \cup [n-1]$, Asm. 2 is satisfied with $R = 1$.*

**Proof.** Using the facts $\|C_{r,\lambda}^{-1}\|_\infty \le 1/\lambda$, $\|X_{n-r}\|_\infty \le \|X_{n-r}\| \le \sqrt{n-r}$ (since we are assuming $\mathcal{X} \subseteq \mathcal{B}_1$) and the definition of $\bar{X}_r$, we have that

$$\|\bar{X}_r\|_\infty = \frac{\lambda}{\sqrt{n-r}}\|X_{n-r}C_{\lambda,r}^{-1}\|_\infty = \frac{\lambda}{\sqrt{n-r}}\|C_{\lambda,r}^{-1}\|_\infty\|X_{n-r}\|_\infty \le 1.$$

We remark that the above steps hold also for the extreme case $r = 0$, according to the associated definitions in that case. ∎

In the case of Ex. 1, we manage to get an estimate of the constant $\sigma$ introduced in Asm. 3. This is reported in the following lemma.

**Lemma 19** (Asm. 3 for $\mathcal{E}_r$ and Ex. 1). *Consider the setting described in the paper and outlined in Eq. (49). Let $\mathcal{X} \subseteq \mathcal{B}_1$ and let the environment satisfy the assumptions in Ex. 1. Then, for any $r \in \{0\} \cup [n-1]$ and for any $\lambda > 0$, Asm. 3 is satisfied with*

$$\sigma_r^2 = \sigma_w^2 + \Big(\frac{1}{n-r} + \frac{r}{(n\lambda)^2}\Big)\sigma_\epsilon^2.$$

**Proof.** We wish to estimate $\sigma_r$ such that

$$\mathbb{E}\Big[\bar{X}_r^\top \bar{\boldsymbol{\epsilon}}_r \bar{\boldsymbol{\epsilon}}_r^\top \bar{X}_r\Big] \preceq \sigma_r^2 \mathbb{E}\Big[\bar{X}_r^\top \bar{X}_r\Big].$$

Since in the setting of Ex. 1 $h_r^* = \bar{w}$ for any $r \in \{0\} \cup [n-1]$, the residuals are given by

$$\bar{\boldsymbol{\epsilon}}_r = \bar{\mathbf{y}}_r - \bar{X}_r \bar{w} = \bar{X}_r v + \frac{1}{\sqrt{n-r}}\Big(\boldsymbol{\epsilon}_{n-r} - \frac{\sqrt{n-r}}{\lambda}\bar{X}_r \frac{X_r^\top \boldsymbol{\epsilon}_r}{n}\Big),$$

with $v = w - \bar{w}$. Hence, we have that

$$\bar{X}_r^\top \bar{\boldsymbol{\epsilon}}_r \bar{\boldsymbol{\epsilon}}_r^\top \bar{X}_r = \bar{X}_r^\top \bar{X}_r v v^\top \bar{X}_r^\top \bar{X}_r + \frac{1}{n-r}\bar{X}_r^\top \boldsymbol{\epsilon}_{n-r}\boldsymbol{\epsilon}_{n-r}^\top \bar{X}_r + \frac{1}{(n\lambda)^2}\bar{X}_r^\top \bar{X}_r X_r^\top \boldsymbol{\epsilon}_r \boldsymbol{\epsilon}_r^\top X_r \bar{X}_r^\top \bar{X}_r$$

$$+ \underbrace{\frac{1}{\sqrt{n-r}}\bar{X}_r^\top \bar{X}_r v \boldsymbol{\epsilon}_{n-r}^\top \bar{X}_r}_{1} - \underbrace{\frac{1}{n\lambda}\bar{X}_r^\top \bar{X}_r v \boldsymbol{\epsilon}_r^\top X_r \bar{X}_r^\top \bar{X}_r}_{2}$$

$$+ \underbrace{\frac{1}{\sqrt{n-r}}\bar{X}_r^\top \boldsymbol{\epsilon}_{n-r} v^\top \bar{X}_r^\top \bar{X}_r}_{3} - \underbrace{\frac{1}{n\lambda\sqrt{n-r}}\bar{X}_r^\top \boldsymbol{\epsilon}_{n-r}\boldsymbol{\epsilon}_r^\top X_r \bar{X}_r^\top \bar{X}_r}_{4}$$

$$- \underbrace{\frac{1}{n\lambda}\bar{X}_r^\top \bar{X}_r X_r^\top \boldsymbol{\epsilon}_r v^\top \bar{X}_r^\top \bar{X}_r}_{5} - \underbrace{\frac{1}{n\lambda\sqrt{n-r}}\bar{X}_r^\top \bar{X}_r X_r^\top \boldsymbol{\epsilon}_r \boldsymbol{\epsilon}_{n-r}^\top \bar{X}_r}_{6}.$$

Now, we focus on the expectation of the term 1 and we observe that, exploiting the closed form of $\bar{X}_r$ and the fact that $\eta \mid p$ has zero-mean, we can write

$$\mathbb{E}\Big[\bar{X}_r^\top \bar{X}_r v \boldsymbol{\epsilon}_{n-r}^\top \bar{X}_r\Big] = \mathbb{E}\Big[\mathbb{E}\Big[\bar{X}_r^\top \bar{X}_r v \boldsymbol{\epsilon}_{n-r}^\top \bar{X}_r \Big| p, w, X_n\Big]\Big]$$

$$= \mathbb{E}\Big[\bar{X}_r^\top \bar{X}_r v \mathbb{E}\big[\boldsymbol{\epsilon}_{n-r}\big| p, w, X_n\big]^\top \bar{X}_r\Big]$$

$$= \mathbb{E}\Big[\bar{X}_r^\top \bar{X}_r v \mathbb{E}\big[\boldsymbol{\epsilon}_{n-r}\big| p\big]^\top \bar{X}_r\Big] = 0,$$

where in the third equality we have exploited the independence of $\boldsymbol{\epsilon}_{n-r}$ with respect to $w, X_n$, conditioned with respect to $p$ (see Rem. 8). Applying a similar reasoning, it is easy to show that also the terms $2 - 6$ have zero-mean. Consequently, we have that

$$\mathbb{E}\Big[\bar{X}_r^\top \bar{\boldsymbol{\epsilon}}_r \bar{\boldsymbol{\epsilon}}_r^\top \bar{X}_r\Big] = \mathbb{E}\Big[\bar{X}_r^\top \bar{X}_r v v^\top \bar{X}_r^\top \bar{X}_r\Big] + \frac{1}{n-r}\mathbb{E}\Big[\bar{X}_r^\top \boldsymbol{\epsilon}_{n-r}\boldsymbol{\epsilon}_{n-r}^\top \bar{X}_r\Big]$$
$$+ \frac{1}{(n\lambda)^2}\mathbb{E}\Big[\bar{X}_r^\top \bar{X}_r X_r^\top \boldsymbol{\epsilon}_r \boldsymbol{\epsilon}_r^\top X_r \bar{X}_r^\top \bar{X}_r\Big]. \tag{50}$$

We now treat the three terms in a separate way. As regards the first term in Eq. (50), using assumption $iii)$ in Ex. 1 and the independence of $w$ and $X_n$ conditioned with respect to $p$ (see Rem. 8), we have that

$$\mathbb{E}\Big[\bar{X}_r^\top \bar{X}_r v v^\top \bar{X}_r^\top \bar{X}_r\Big] = \mathbb{E}\Big[\mathbb{E}\Big[\bar{X}_r^\top \bar{X}_r v v^\top \bar{X}_r^\top \bar{X}_r \Big| p, X_n\Big]\Big]$$
$$= \mathbb{E}\Big[\bar{X}_r^\top \bar{X}_r \mathbb{E}\big[v v^\top \big| p, X_n\big] \bar{X}_r^\top \bar{X}_r\Big]$$
$$= \mathbb{E}\Big[\bar{X}_r^\top \bar{X}_r \mathbb{E}\big[v v^\top \big| p\big] \bar{X}_r^\top \bar{X}_r\Big]$$
$$\preceq \sigma_w^2 \mathbb{E}\Big[\bar{X}_r^\top \bar{X}_r \bar{X}_r^\top \bar{X}_r\Big]$$
$$\preceq \sigma_w^2 \mathbb{E}\Big[\bar{X}_r^\top \bar{X}_r\Big],$$

where, in the first inequality we have applied Lemma 4 with

$$U = \sigma_w^2 I - \mathbb{E}\big[v v^\top \big| p\big] \qquad V = \bar{X}_r^\top \bar{X}_r,$$

and in the second inequality we have applied Cor. 5 with $W = \bar{X}_r^\top \bar{X}_r$ and Lemma 18, according to which $\|\bar{X}_r^\top \bar{X}_r\|_\infty = \|\bar{X}_r\|_\infty^2 \leq 1$. As regards the second term in Eq. (50), using assumption $i)$ in Ex. 1 and the independence of the points in the datasets, we have that $\mathbb{E}\big[\boldsymbol{\epsilon}_{n-r}\boldsymbol{\epsilon}_{n-r}^\top \big| p\big] \preceq \sigma_\epsilon^2 I$, consequently, exploiting the independence of $\boldsymbol{\epsilon}_{n-r}$ and $X_n$ conditioned with respect to $p$ (see Rem. 8), we can write

$$\frac{1}{n-r}\mathbb{E}\Big[\bar{X}_r^\top \boldsymbol{\epsilon}_{n-r}\boldsymbol{\epsilon}_{n-r}^\top \bar{X}_r\Big] = \frac{1}{n-r}\mathbb{E}\Big[\mathbb{E}\big[\bar{X}_r^\top \boldsymbol{\epsilon}_{n-r}\boldsymbol{\epsilon}_{n-r}^\top \bar{X}_r \big| p, X_n\big]\Big]$$
$$= \frac{1}{n-r}\mathbb{E}\Big[\bar{X}_r^\top \mathbb{E}\big[\boldsymbol{\epsilon}_{n-r}\boldsymbol{\epsilon}_{n-r}^\top \big| p, X_n\big] \bar{X}_r\Big]$$
$$= \frac{1}{n-r}\mathbb{E}\Big[\bar{X}_r^\top \mathbb{E}\big[\boldsymbol{\epsilon}_{n-r}\boldsymbol{\epsilon}_{n-r}^\top \big| p\big] \bar{X}_r\Big]$$
$$\preceq \frac{\sigma_\epsilon^2}{n-r}\mathbb{E}\Big[\bar{X}_r^\top \bar{X}_r\Big],$$

where in the last step we have applied Lemma 4 with

$$U = \sigma_\epsilon^2 I - \mathbb{E}\big[\boldsymbol{\epsilon}_{n-r}\boldsymbol{\epsilon}_{n-r}^\top \big| p\big] \qquad V = \bar{X}_r.$$

As regards the third term in Eq. (50), we start observing that, using again assumption $i)$ in Ex. 1 and the independence of the points in the datasets, we have that $\big[\boldsymbol{\epsilon}_r \boldsymbol{\epsilon}_r^\top \big| p\big] \preceq \sigma_\epsilon^2 I$. Hence, exploiting the independence of $\boldsymbol{\epsilon}_r$ and $X_n$ conditioned with respect to $p$ (see Rem. 8), we can write

$$\frac{1}{(n\lambda)^2}\mathbb{E}\Big[\bar{X}_r^\top \bar{X}_r X_r^\top \boldsymbol{\epsilon}_r \boldsymbol{\epsilon}_r^\top X_r \bar{X}_r^\top \bar{X}_r\Big] = \frac{1}{(n\lambda)^2}\mathbb{E}\Big[\mathbb{E}\big[\bar{X}_r^\top \bar{X}_r X_r^\top \boldsymbol{\epsilon}_r \boldsymbol{\epsilon}_r^\top X_r \bar{X}_r^\top \bar{X}_r \big| p, X_n\big]\Big]$$
$$= \frac{1}{(n\lambda)^2}\mathbb{E}\Big[\bar{X}_r^\top \bar{X}_r X_r^\top \mathbb{E}\big[\boldsymbol{\epsilon}_r \boldsymbol{\epsilon}_r^\top \big| p, X_n\big] X_r \bar{X}_r^\top \bar{X}_r\Big]$$
$$= \frac{1}{(n\lambda)^2}\mathbb{E}\Big[\bar{X}_r^\top \bar{X}_r X_r^\top \mathbb{E}\big[\boldsymbol{\epsilon}_r \boldsymbol{\epsilon}_r^\top \big| p\big] X_r \bar{X}_r^\top \bar{X}_r\Big]$$
$$\preceq \frac{\sigma_\epsilon^2}{(n\lambda)^2}\mathbb{E}\Big[\bar{X}_r^\top \bar{X}_r X_r^\top X_r \bar{X}_r^\top \bar{X}_r\Big]$$
$$\preceq \frac{\sigma_\epsilon^2 r}{(n\lambda)^2}\mathbb{E}\Big[\bar{X}_r^\top \bar{X}_r \bar{X}_r^\top \bar{X}_r\Big]$$
$$\preceq \frac{\sigma_\epsilon^2 r}{(n\lambda)^2}\mathbb{E}\Big[\bar{X}_r^\top \bar{X}_r\Big],$$

where, in the first inequality we have applied Lemma 4 with

$$U = \sigma_\epsilon^2 I - \mathbb{E}\big[\boldsymbol{\epsilon}_r \boldsymbol{\epsilon}_r^\top | p\big] \qquad V = X_r \bar{X}_r^\top \bar{X}_r$$

and in the second inequality we have applied Lemma 4 with

$$U = \big\|X_r^\top X_r\big\|_\infty I - X_r^\top X_r \qquad V = \bar{X}_r^\top \bar{X}_r$$

and the fact that $\|X_r^\top X_r\|_\infty \leq \|X_r^\top X_r\| \leq r$ (since we are assuming $\mathcal{X} \subseteq \mathcal{B}_1$). Finally, in the third inequality, we have applied Cor. 5 with $W = \bar{X}_r^\top \bar{X}_r$ and Lemma 18, according to which $\|\bar{X}_r^\top \bar{X}_r\|_\infty = \|\bar{X}_r\|^2 \leq 1$. Putting all together, we conclude that

$$\mathbb{E}\Big[\bar{X}_r^\top \bar{\boldsymbol{\epsilon}}_r \bar{\boldsymbol{\epsilon}}_r^\top \bar{X}_r\Big] \preceq \Big(\sigma_w^2 + \Big(\frac{1}{n-r} + \frac{r}{(n\lambda)^2}\Big)\sigma_\epsilon^2\Big)\mathbb{E}\Big[\bar{X}_r^\top \bar{X}_r\Big].$$

We conclude observing that the above steps hold also for the extreme case $r = 0$, according to the associated definitions in that case. ∎

We observe that the upper bound given in Lemma 19 is decreasing in $r$. This confirms in some way what we have already observed in Rem. 5 in the paper: in the specific setting of Ex. 1, using more than one test point in the definition of the function $\mathcal{E}_r$ has the same effect of traditional mini-batches, i.e. it reduces the variance on the unbiased estimates of the true gradient computed at the true solution. We now are ready to state the adaptation of Thm. 12 for the output of Alg. 1 introduced in the paper, in the setting of Ex. 1.

**Theorem 20** (Convergence rate of SGD to $\mathcal{E}_r$ for Ex. 1)**.** *Assume $\mathcal{X} \subseteq \mathcal{B}_1$ and let $\bar{h}_{T,r,\lambda}$ be the output of Alg. 1, for any $r \in \{0\} \cup [n-1]$ and $\lambda > 0$. Then, the expected excess transfer risk of the algorithm in Eqs. (8)-(9) with parameter $\bar{h}_{T,r,\lambda}$ trained with $r$ points over the environment in Ex. 1 is bounded by*

$$\mathbb{E}\big[\mathcal{E}_r(\bar{h}_{T,r,\lambda}) - \mathcal{E}_r(h_r^*)\big] \leq \frac{2\big\|h^{(0)} - \bar{w}\big\|^2}{\gamma(T+1)} + \Big(\sigma_w^2 + \Big(\frac{1}{n-r} + \frac{r}{(n\lambda)^2}\Big)\sigma_\epsilon^2\Big)\frac{2d}{T+1},$$

*where the expectation is over the datasets $Z^{(1)}, \dots, Z^{(T)}$.*

**Proof.** The statement immediately follows from applying Thm. 12 to the context outlined in Eq. (49) in the setting of Ex. 1. We choose the upper bound on the step size $\gamma$ in Asm. 2 according to Lemma 18 and we use the estimate for $\sigma^2$ in Asm. 3 obtained in Lemma 19. Finally, for Ex. 1, we know that, for any $r \in \{0\} \cup [n-1]$, $h_r^*$ coincides with the mean $\bar{w}$ of the environment. ∎

# E  Proof of Prop. 3

In this section we give the proof of Prop. 3.

**Proposition 3.** *Assume $\mathcal{X} \subseteq \mathcal{B}_1$ and, for any $r \in \{0\} \cup [n-1]$ and $\lambda > 0$, let $\bar{h}_{T,r,\lambda}$ be the output of Alg. 1. Then, the expected excess transfer risk of the algorithm in Eqs. (8)-(9) with parameter $\bar{h}_{T,r,\lambda}$ trained with $n$ points over the environment in Ex. 1 is bounded by*

$$\mathbb{E}\big[\mathcal{E}_n(\bar{h}_{T,r,\lambda}) - \mathcal{E}_n(h_n^*)\big] \leq \frac{(r/n+\lambda)^2}{\lambda^2}\frac{4K_\rho}{T+1}\Big(\frac{1}{\gamma}\big\|h^{(0)} - \bar{w}\big\|^2 + \Big(\sigma_w^2 + \Big(\frac{1}{n-r} + \frac{r}{(n\lambda)^2}\Big)\sigma_\epsilon^2\Big)d\Big),$$

*where the expectation is over the datasets $Z_n^{(1)}, \dots, Z_n^{(T)}$ and $K_\rho$ is condition number of the environment defined as*

$$K_\rho = \frac{\mathbb{E}_{\mu \sim \rho}\big\|\Sigma_\mu\big\|_\infty}{\mathbb{E}_{\mu \sim \rho}\lambda_{\min}\big(\Sigma_\mu\big)}. \tag{13}$$

**Proof.** As described in the proof sketch in Sec. 4, in order to prove the statement, we start from the decomposition

$$\mathbb{E}\big[\mathcal{E}_n(\bar{h}_{T,r,\lambda}) - \mathcal{E}_n(h_n^*)\big] \leq \underbrace{\mathbb{E}\Big[\big\|\bar{\Sigma}_n^{1/2}\big(\bar{h}_{T,r,\lambda} - h_r^*\big)\big\|^2\Big]}_{A} + \underbrace{\mathbb{E}\Big[\big\|\bar{\Sigma}_n^{1/2}\big(h_r^* - h_n^*\big)\big\|^2\Big]}_{B}.$$

The proof of the proposition proceeds by bounding the two separate terms $A$ and $B$. Thanks to the structure of the environment under consideration, we have that $h_r^* = \bar{w}$ for any $r$ (see Ex. 1), so, the term $B$ vanishes. To bound the term $A$ we observe that

$$A = \mathbb{E}\Big[\big\|\bar{\Sigma}_n^{1/2}(\bar{h}_{T,r,\lambda} - h_r^*)\big\|^2\Big] \le \big\|\bar{\Sigma}_n\big\|_\infty \mathbb{E}\Big[\big\|\bar{h}_{T,r,\lambda} - h_r^*\big\|^2\Big] \tag{51}$$

and

$$\lambda_{\min}(\bar{\Sigma}_r)\mathbb{E}\Big[\big\|\bar{h}_{T,r,\lambda} - h_r^*\big\|^2\Big] \le \mathbb{E}\Big[\big\|\bar{\Sigma}_r^{1/2}(\bar{h}_{T,r,\lambda} - h_r^*)\big\|^2\Big]. \tag{52}$$

Consequently, since the matrix $\bar{\Sigma}_r$ is invertible (see Prop. 2), combining Eq. (51) with Eq. (52) and exploiting the LS structure of the function $\mathcal{E}_r$ (see Prop. 2), we can write

$$A \le \frac{2\|\bar{\Sigma}_n\|_\infty}{\lambda_{\min}(\bar{\Sigma}_r)}\frac{1}{2}\mathbb{E}\Big[\big\|\bar{\Sigma}_r^{1/2}(\bar{h}_{T,r,\lambda} - h_r^*)\big\|^2\Big] = \frac{2\|\bar{\Sigma}_n\|_\infty}{\lambda_{\min}(\bar{\Sigma}_r)}\mathbb{E}\big[\mathcal{E}_r(\bar{h}_{T,r,\lambda}) - \mathcal{E}_r(h_r^*)\big]. \tag{53}$$

Now, introducing the condition number of the environment

$$K_\rho = \frac{\mathbb{E}_{\mu\sim\rho}\|\Sigma_\mu\|_\infty}{\mathbb{E}_{\mu\sim\rho}\lambda_{\min}(\Sigma_\mu)}$$

and exploiting Lemma 11, we can write

$$\frac{\|\bar{\Sigma}_n\|_\infty}{\lambda_{\min}(\bar{\Sigma}_r)} \le \frac{K_\rho(r/n + \lambda)^2}{\lambda^2}.$$

Consequently, coming back to Eq. (53), we have

$$A \le \frac{2K_\rho(r/n + \lambda)^2}{\lambda^2}\mathbb{E}\big[\mathcal{E}_r(\bar{h}_{T,r,\lambda}) - \mathcal{E}_r(h_r^*)\big].$$

Using the bound given in Thm. 20 for the term $\mathbb{E}\big[\mathcal{E}_r(\bar{h}_{T,r,\lambda}) - \mathcal{E}_r(h_r^*)\big]$, we get the final statement of the proposition. ∎

## F  How to tune the hyper-parameters in our LTL setting

Denote by $\bar{h}_{T,r,\lambda}$ the output of Alg. 1 computed with $T$ iterations (hence $T$ tasks) with values $r$ and $\lambda$. In all experiments, we obtain this estimator $\bar{h}_{T_{\mathrm{tr}},r,\lambda}$ by learning it on a dataset $\mathbf{Z}_{\mathrm{tr}}$ of $T_{\mathrm{tr}}$ *training* tasks, each comprising a dataset $Z_n$ of $n$ input-output pairs $(x, y) \in \mathcal{X} \times \mathcal{Y}$. During the training phase, each of these datasets is splitted into two parts $Z_n = (Z_r, Z_{n-r})$ and we apply the procedure described in the paper in Alg. 1 in order to compute the estimator. We perform this meta-training for different values of $\lambda \in \{\lambda_1, \dots, \lambda_p\}$ and $r \in \{0\} \cup [n-1]$ and we select the best estimator based on the prediction error measured on a separate set $\mathbf{Z}_{\mathrm{va}}$ of $T_{\mathrm{va}}$ *validation* tasks. Once such optimal $\lambda$ and $r$ values have been selected, we report the generalization performance of the corresponding estimator on a set $\mathbf{Z}_{\mathrm{te}}$ of $T_{\mathrm{te}}$ *test* tasks. The tasks in the test and validation sets $\mathbf{Z}_{\mathrm{te}}$ and $\mathbf{Z}_{\mathrm{va}}$ are all provided with both a training and test dataset both sampled from the same distribution. Note that, since we are interested in measuring the performance of the algorithm trained with $n$ points (as already stressed in the paper we aim at minimizing $\mathcal{E}_n$ and not $\mathcal{E}_r$), the training datasets have all the same sample size $n$ as those in the meta-training datasets in $\mathbf{Z}_{\mathrm{tr}}$, while the test datasets contain $n'$ points each, for some positive integer $n'$. Indeed, in order to evaluate the performance of a bias $h$, we need to first train the corresponding algorithm $w_h$ on the training dataset $Z_n$, and then test its performance on the test set $Z'_{n'}$, by computing the empirical risk $\frac{1}{2n'}\big\|X'_{n'}w_h(Z_n) - \mathbf{y}'_{n'}\big\|^2$. For instance, for the synthetic experiments reported in the paper, we chose $n = 20$ and $n' = 100$. Finally, since we are considering the online setting, the training datasets arrive one at the time, therefore model selection is performed *online*: the system keeps track of all candidate values $\bar{h}_{T_{\mathrm{tr}},r,\lambda_j}$, $r \in \{0\} \cup [n-1]$, $j \in [p]$, and, whenever a new training task is presented, these vectors are all updated by incorporating the corresponding new observations. The best bias $h$ is then returned at each iteration, based on its performance on the validation set $\mathbf{Z}_{\mathrm{va}}$. The previous procedure describes how to tune simultaneously both $\lambda$ and $r$. In some experiments reported in the paper, we just tuned one of them; in such a case the procedure is analogous to that described above.

**Note on Real Data**. In real settings, the assumptions of having the same number $n$ of training points available for each task $t$ is often restrictive. Indeed, on the School dataset considered in this work, datasets can have a very different number $n_t$ of training points. Hence, in order to validate the theoretical analysis reported in this paper, in our experiments we down-sampled the datasets with more training examples in order to maintain the same $n$ across all the tasks. However, it is natural to expect that leveraging more training points when available should be more favorable in terms of overall performance. This is a key question that introduces complications to our analysis and that we plan to investigate in the future. We refer to the code for more details.