[Reviews · NeurIPS 2018]

Reviewer 1



The paper considers a learning to learn problem (LTL) where the tasks are linear regression and ridge regression around a common mean as a family of algorithms. Unlike in the previous works where the empirical risk of the new task is considered, the author(s) proposed to learn the common mean by directly minimizing the transfer risk. The paper uses SGD to incrementally learn the new tasks in LTL setting using the online splitting approach and studied its statistical guarantees (under certain assumptions). The problem is interesting and as the author mentioned in the paper there are only very few papers with the theoretical guarantees for LTL. The paper is well organized and easy to follow. Even though the underlying algorithm has been considered already (Evgeniou et. Al., KDD 2004), the application of this approach to LTL where the common mean (h) is learned by the meta algorithm using simple transformation while the inner algorithm (that depends on h) works well on new (but similar) tasks. Under the assumption of example 1, the optimal splitting (r=0) results in using the common mean (h) for the new tasks. It will be interesting to see how this can be related to the lifelong learning with weighted majority voting (Pentina and Urner NIPS 2016) where the weights will be learned by the meta algorithm. Usage of dataset splitting for learning this common mean is novel and the transfer risk bound shows how the splitting of the dataset is affected under the strong assumption E[w|p]=\bar{w}. It is unclear from the bound in Prop 3 for Example 1, that we can get a consistent estimator as T-> \infty even when the sample size is kept constant. This doesn’t seem directly applicable for the environment and assumptions considered in the Example 2. This paper has several important theoretical contributions to the LTL problem setting and I recommend the acceptance of this paper.

Reviewer 2



This paper addresses the question of learning to learn (LTL) in the situation where the datasets come in streams. The model assumes a common mean vector that has to be learning throughout the process. To this end, each training dataset is split into a training and a test dataset. The dataset is used to learn the common mean, using a stochastic gradient method. Theoretical results on the excess risk are provided. The paper is well-written: - organization is good - depiction of the algorithm is precise and concise - theoretical guarantees are provided - empirical evaluation (even though more results could be provided) is provided. This work raises three questions: - instead of a stochastic gradient method, wouldn't have it been possible to implement a passive-aggressive (Crammer and Singer) scheme. It seems it might be appropriate to carry over the idea of having a common mean between tasks, and it could have helped provide regret-like results on the training performance (in short, just to be clearer, such a strategy would consist in seeing each dataset as a training instance); - also: what about using fast Loo error estimation for the stochastic gradient strategy in the present ridge-regression-case? This would help having an algorithm independent of $r$; - how is it difficult to state the results for other kinds of inner training procedures, e.g. SVM? Overall, a very interesting paper. Typos ==== will be useD sacrify -> sacrifice returns always -> always returns an important aspect -> an important feature bring similar -> brings similar in a such case -> in such a case which will addressed -> which will be addressed As regards the term -> Regarding the term where we K_\rho -> where K_\rho investing empirically -> empirically investing we have presentED which highlightS

Reviewer 3



This paper studies how to define an algorithm that, given an increasing number of tasks sampled from a defined environment, will train on them and learn a model that will be well suited for any new task sampled from the same environment. The scenario just described corresponds to the 'learning to lean' problem where a learning agent improves its learning performance with the number of tasks. Specifically in this work the focus is on the 'ridge regression' family of algorithms and the environment consists in tasks that can be solved by ridge regression with models around a common mean. In other words, we need a learning algorithm that besides solving regression problems, progressively learns how to approximate the environment model mean. The transfer risk is a measure of how much the knowledge acquired over certain available tasks allow to improve future learning. In the proposed setting the transfer risk corresponds to the difference between the estimated model environmental mean and the correct environmental mean. It can be well estimated with theoretical guarantees and can be explicitly minimized to 'learn to learn' the optimal regression model. The novel contribution of the paper is in the choice of splitting the data of each tasks in two parts, one on which the regression problem is learned and the training error is minimized and a second one useful to estimate and minimize the transfer risk. Finally the proposed strategy is tested onto two synthetic dataset where the results confirm the paper thesis. Ovearall the paper is well written but I find it dense in some points which makes it a little bit difficult to read. The proposed method is theoretically grounded and I did not find any evident flaw. I just remain with one doubt: what could be real-world cases where the assumptions done for the synthetic settings hold? And what would be the challenges to be tackled in case of extensions to non-convex learning problems? After rebuttal ----------------- I've noticed that in the rebuttal some of the reviewers' questions are indicated as 'open problems' or deferred to future works, Anyway I do not have strong arguments against this paper. I align with the other reviewers's score by raising my rate from 6 to 7.

Reviewer 4



The authors consider the learning-to-learn setting under the assumption that all the tasks are linear regression problems and their solutions are close to each other in l2 norm. First, the authors observe that the transfer risk in case of squared loss and ridge regression is basically a normal expected squared loss in a modified space. Second, the authors propose to split the data for every task in two subsets to imitate the train/test split and re-write the objective function according to this data processing. Lastly, they observe that receiving one task at a time allows them to do SGD on that modified objective. Implications of Proposition 3 are quite interesting and, I think, deserve a more elaborate discussion in the manuscript. First, it suggests to not perform any task-specific training (r=0) and implies that learning is possible even when the learner observes only one sample per task(n=1) . This feature of the considered setting is quite interesting. May be it is connected to the fact that there is no clear advantage of having more data in learning algorithm (3) (even in case of infinite training sets it will not find the optimal weight vector). Also, the number of samples per task n influences only the term corresponding to the noise-variance, while all other terms decay only with T->infinity.